# Proteome-scale characterisation of motif-based interactome rewiring by disease mutations

Johanna Kliche [ID][1], Leandro Simonetti [ID][1], Izabella Krystkowiak [ID][2], Hanna Kuss [ID][1,4], Marcel Diallo [ID][3], Emma Rask[1], Jakob Nilsson [ID][3], Norman E Davey [ID][2✉] & Ylva Ivarsson [ID][1✉]

## Abstract

**Whole genome and exome sequencing are reporting on hundreds of thousands of missense mutations. Taking a pan-disease approach, we explored how mutations in intrinsically disordered regions (IDRs) break or generate protein interactions mediated by short linear motifs. We created a peptide-phage display library tiling ~57,000 peptides from the IDRs of the human proteome overlapping 12,301 single nucleotide variants associated with diverse phenotypes including cancer, metabolic diseases and neurological diseases. By screening 80 human proteins, we identified 366 mutation-modulated interactions, with half of the mutations diminishing binding, and half enhancing binding or creating novel interaction interfaces. The effects of the mutations were confirmed by affinity measurements. In cellular assays, the effects of motif-disruptive mutations were validated, including loss of a nuclear localisation signal in the cell division control protein CDC45 by a mutation associated with Meier-Gorlin syndrome. The study provides insights into how disease-associated mutations may perturb and rewire the motif-based interactome.**

**Keywords** Genetic Variation; Protein–Protein Interaction; Short Linear Motif; Phage Display; CDC45
**Subject Categories** Chromatin, Transcription & Genomics; Proteomics

## Introduction

Large-scale sequencing approaches have provided the scientific community with detailed variation information of the human genome, represented by both non-pathogenic polymorphisms as well as rare and common disease-causing mutations (Cooper and Shendure, 2011; Bamshad et al, 2011). Approximately 50% of the disease-causing mutations are represented by non-synonymous single nucleotide variants (SNVs), which are characterised by changes in one nucleotide resulting in an amino acid substitution at the protein level (Bamshad et al, 2011). These mutations may result in protein dysfunction that can be manifested on multiple levels, such as activity, stability, conformation and protein–protein interactions (PPIs) (Stefl et al, 2013). Thus, it is crucial to shed light on the functional consequences elicited by non-synonymous SNVs on the protein level.

Numerous computational tools have been developed to predict the impact of non-synonymous SNVs in human disease. These tools mainly focus on the effect of SNVs on the function and folding of structured parts of the proteome (Stitziel et al, 2004; Shihab et al, 2014; Lopes et al, 2012; Hassan et al, 2019). Many disease-causing mutations occur in the core of folded regions of proteins, affecting their structural stability (Wang and Moult, 2001). Other mutations are found on the protein surface and often occur in PPI interfaces (David et al, 2012; David and Sternberg, 2015; Wong et al, 2020). Approximately 10% of missense SNVs have been suggested to have a disruptive impact on PPIs (Fragoza et al, 2019). Furthermore, ~30% of disease mutations were reported to have edgetic effects, perturbing some but not all interactions of a given protein (Sahni et al, 2015). The establishment of onco-specific PPI networks correlating with patient survival, and that both somatic and germline mutations associated with diseases more frequently target protein binding interfaces as compared to non-interface positions, further emphasises that disease-associated mutations often perturb PPI networks (Cheng et al, 2021).

About 40% of the human proteome is fully or partially intrinsically disordered, and about 20% of the disease-associated missense mutations map to these regions (Vacic et al, 2012; Uyar et al, 2014; Mészáros et al, 2021). Intrinsically disordered regions (IDRs) harbour most of the short linear motifs (SLiMs) of the proteome. SLiMs are compact interaction modules, typically encoded by 3–10 amino acid long stretches (Davey et al, 2012; Tompa et al, 2014). SLiM-based interactions are critical for cellular function, which is exemplified by their major role in cell signalling, regulation of transcription and cell cycle progression (Wright and Dyson, 2015). Mutations of SLiMs are associated with various diseases, including cancer and neurological diseases, and disease-related mutations occur more often in SLiMs than neutral

[1]Department of Chemistry - BMC, Box 576, Husargatan 3, 751 23 Uppsala, Sweden. [2]Division of Cancer Biology, Institute of Cancer Research, Chester Beatty Laboratories, 237 Fulham Road, SW3 6JB, Chelsea, London, UK. [3]Novo Nordisk Foundation Center for Protein Research, Faculty of Health and Medical Sciences, University of Copenhagen, Blegdamsvej 3B, 2200 Copenhagen, Denmark. [4]Present address: University of Münster, Institute of Pharmaceutical and Medicinal Chemistry, DE-48149 Münster, Germany. ✉E-mail: norman.davey@icr.ac.uk; ylva.ivarsson@kemi.uu.se

mutations (Uversky et al, 2008; Kulkarni and Uversky, 2019; Uyar et al, 2014). For example, mutation of degradation motifs (degrons) that target proteins for E3 ligase-dependent ubiquitination and proteasomal degradation have been associated with cancer (Mészáros et al, 2017). A prominent example is the loss-of-function mutations in the β-catenin (CTNNB1) degron, which acts as a driver of oncogenesis (Provost et al, 2005). Although less reported, disease-associated mutations can generate novel SLiMs, and thereby rewire the interaction network. Such neo-interactions were found to be created by mutations generating dileucine motifs in the cytosolic IDRs of transmembrane proteins, which results in increased clathrin-mediated endocytosis and consequently in mislocalisation of the proteins (Meyer et al, 2018). Together this supports the notion that disease-associated mutations within IDRs can profoundly contribute to disease progression by affecting the interaction landscape.

In this study, we apply mutational proteomic peptide-phage display (ProP-PD) for large-scale identification of genetic variations within IDRs resulting in altered SLiM-based PPIs. The approach is a variation of conventional ProP-PD, where the p8 coat protein of the M13 bacteriophage is engineered to display the IDRs of a chosen proteome. By using the ProP-PD library in selections against purified bait proteins, we identify interaction partners and delineate binding peptides and motifs (Benz et al, 2022). In mutational phage display, each peptide is represented by two versions in the library, one wild-type and one mutant peptide. In this fashion, it is possible to discriminate the binding preferences for either of the two versions of the peptide in a single experiment, while screening thousands of peptide pairs at the same time. The approach is conceptually similar to phosphomimetic ProP-PD, in which phosphomimetic versions of the peptides are used to delineate potential phospho-modulated binding of the bait protein domains (Sundell et al, 2018; Kliche et al, 2023).

We developed a novel phage library, the genetic variation HD2 library (GenVar_HD2), covering non-synonymous SNVs from the IDRs of the human proteome to screen for interactome changes caused by mutations on a large-scale. We performed parallel selections of 80 bait protein domains against the GenVar_HD2 library and identified 275 mutations associated with a broad spectrum of diseases, from cancer to neurological disorders. We discovered mutations that abrogate, weaken, enhance or de novo generate SLiM-based PPIs (366 domain-mutation pairs). Among the interactions, we uncover a novel nuclear localisation signal (NLS) in the Cell division control protein 45 (CDC45) that binds importin subunit α-8 (KPNA7), and is disrupted by a R157C mutation associated with a developmental disorder called Meier-Gorlin syndrome. In summary, we provide a panoramic overview of how the SLiM-mediated interactome is perturbed as a consequence of genetic variations, and offer molecular insights that decode the link between disease and mutations in the IDRs of the human proteome.

# Results

## Design and generation of the GenVar_HD2 library

We generated a mutational ProP-PD library (GenVar_HD2) that combines the IDRs of the human proteome as defined previously

(Benz et al, 2022) with non-synonymous SNVs to probe for changes in motif-based PPIs caused by the mutations (Fig. 1A). Pathogenic, or likely pathogenic, SNVs were retrieved from several databases assembling information on human genetic variation such as ClinVar (Landrum and Kattman, 2018), the Genome Aggregation Database (gnomAD) (Gudmundsson et al, 2021), the Cancer Genome Atlas (TCGA) (Hutter and Zenklusen, 2018) and UniProt Human polymorphisms and disease mutations (McGarvey et al, 2019) (see Method section for full details). The peptide-phage display library was designed to tile each mutational site with both wild-type and mutant peptides, and the final GenVar_HD2 library design contains a total of 12,301 mutations found in 1915 prey proteins covered by 56,911 unique peptides. These represent 20,434 wild-type peptides and 36,479 wild-type/mutant peptide pairs, as, for a subset of the design, a single wild-type peptide has multiple corresponding mutated variants (Fig. 1B; Dataset EV1A–C). Of the included mutations, 11,305 (91.9%) were classified as pathogenic, the majority of which represent somatic mutations (10,085 mutations; 82.0%) (Fig. 1C; Dataset EV1C). The mutations map back to numerous disease conditions, which were categorised into 11 broad disease classes, including neurological, cancer and metabolic diseases. Notably, 22.9% of the mutations belong to more than one disease category (mixed categorisation, e.g. mutations belonging to both metabolic diseases and cancer) (Fig. 1D; Dataset EV1D). An oligonucleotide pool encoding the designed peptides was obtained and the GenVar_HD2 library was generated by genetically fusing the obtained oligonucleotide library to a phagemid vector encoding the major coat protein p8 of the M13 bacteriophages. This phagemid pool was used to generate a peptide-phage library and next-generation sequencing (NGS) confirmed that at least 94% of the mutation sites are covered by at least one complete wild-type/mutant peptide pair (Appendix Fig. S1A–D). Sequencing further indicated that at least 87.4% of the designed peptides were present in the physical phage library (Appendix Fig. S1A) and that wild-type and mutant peptide pairs were equally represented in the library (Appendix Fig. S1C).

## Bait protein collection and phage selection

We expressed and purified 91 glutathione transferase (GST)- or maltose binding protein (MBP)-tagged fusion proteins to use as baits in phage selections (Dataset EV2A,B). We performed initial selections against the bait protein domains in triplicates, assessed the enrichment of binding phage by phage pool enzyme-linked immunosorbent assay (ELISA) and found 80 of the baits to successfully enrich binding phages in comparison to the negative control proteins (GST, MBP). Additional selections were performed against these baits to ensure at least five replicate selections per bait protein. The final bait collection included domains from proteins with (i) previously reported cancer association, either as cancer-associated genes (Zheng et al, 2021) or as onco-PPIs that have computationally been predicted to be perturbed by disease mutations (Cheng et al, 2021) ($n = 24$), and (ii) proteins for which interactions have been suggested to be disrupted by genetic variation based on large-scale yeast-two-hybrid screening ($n = 11$; (Fragoza et al, 2019)). Seven proteins belonged to both groups. Eleven of the baits within those two groups were previously reported as baits against the HD2 library (Benz et al, 2022), and we included an additional set of baits from this collection (HD2 binder; $n = 8$). Last,

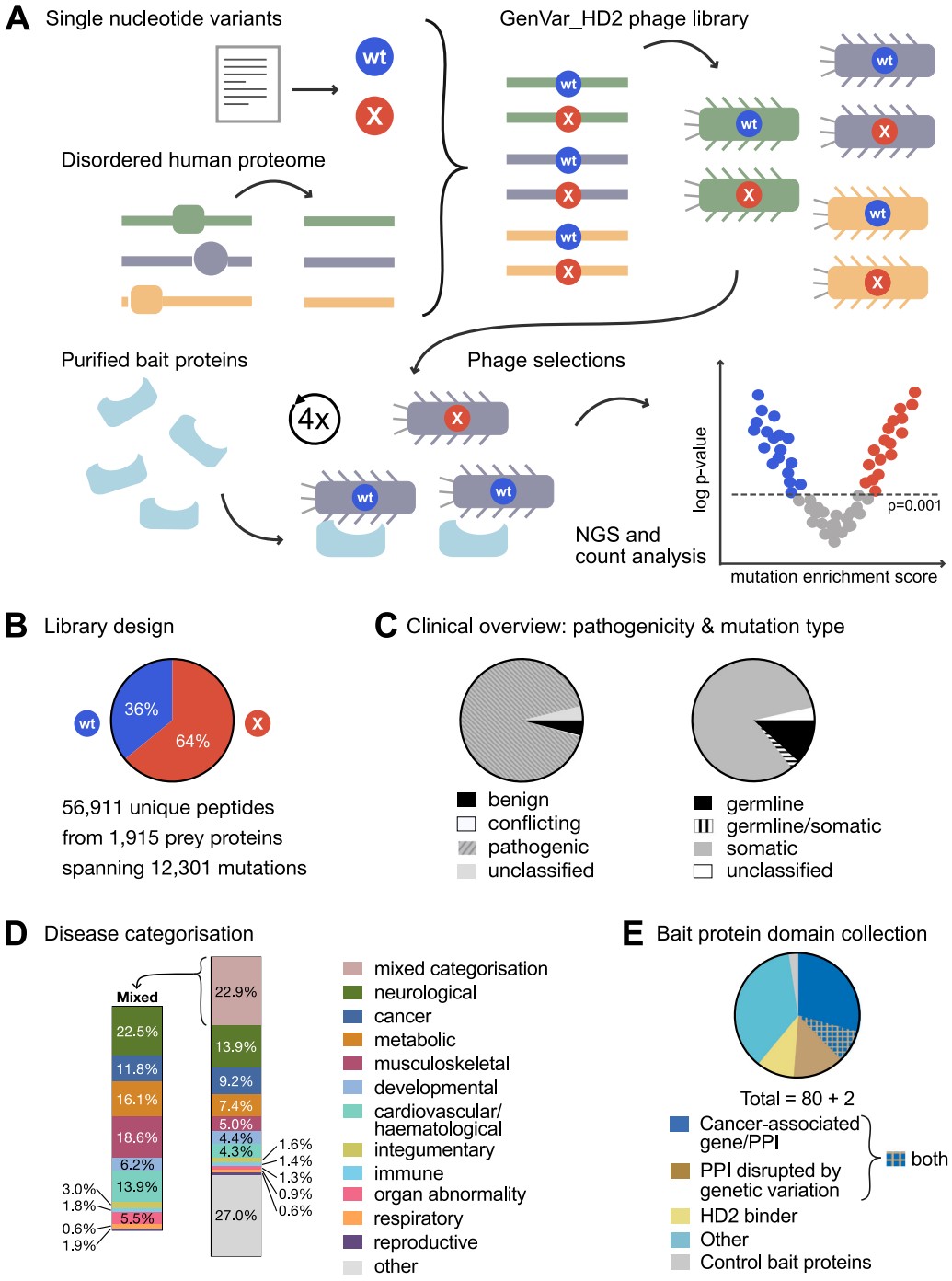

**Figure 1. Study and library design.**

(**A**) Overview of the workflow of the GenVar_HD2 library design, phage selections and data analysis. (**B**) GenVar_HD2 design parameters. (**C**) Mutation distribution by phenotype (left) and by type of mutation (right). (**D**) Distribution of disease mutation categories in the library design. Mutations belonging to more than one disease category (e.g. metabolic and cancer) are categorised as mixed. The distribution of the disease categories in the mixed group is indicated in the left panel. (**E**) Bait protein domain collection and categorisation.

we added domains to increase the variety of peptide binding domain families ($n = 30$) (Fig. 1E; Dataset EV2A,C).

## GenVar_HD2 selections reveal binding-enhancing and -diminishing mutations

After phage selection, the peptide encoding regions of the binding enriched phage pools were barcoded by PCR and analysed by NGS. The GenVar_HD2 ProP-PD selections resulted in 2225 unique medium/high confidence domain-peptide interactions, corresponding to 1317 domain-peptide region interactions and 1230 unique PPIs as the IDRs are tiled by overlapping peptides. Of the PPIs, 123 have been previously reported, and out of those, 37 were also found in our previous study (Benz et al, 2022) (Fig. 2A; Dataset EV3). The phage selection conferred a 2.6-fold enrichment of known binders based on the total number of known binders represented in the library. Furthermore, bait proteins and their found ligands shared gene ontology (GO) terms related to localisation, function and processes to a larger extent than expected by chance (that is, as compared to the background; Appendix Fig. S2; Dataset EV4) when quantified by a classical GO term analysis. The recall of previously reported SLiM-based interactions was 19.9%, which is similar to the previously estimated recall (19.5%) based on selections against the HD2 library. The large number of novel interactions found through the deep phage-based interaction screening (multiple repeats, relatively limited library size) reflects the still sparse coverage of the SLiM-based interactions as well as the generation of neo-interactions by the mutations. For example, we previously screened the NTF domains of the stress-granule-associated proteins G3BP1 and 2 (Ras-GTPase-activating protein-binding proteins 1 and 2) using our HD2 library and our phosphomimetic ProP-PD library and identified 62 interactors for the two protein domains (Kliche et al, 2023; Kruse et al, 2021). Here, we uncover a set of 25 interactions (23 new to this study), of which eight are neo-interactions as discussed later (Dataset EV5).

We delineated the effect of mutations on binding based on the relative NGS counts for wild-type and mutant peptides obtained from the GenVar_HD2 selections using our established analysis pipeline for mutational ProP-PD results (Kliche et al, 2023) (Fig. 1A). In brief, we first collapsed domain-peptide interactions reporting on the same mutation site to retrieve the domain-mutation pairs (1773 pairs, Fig. 2A; Dataset EV6) and the average effect of a given mutation on the binding of the bait protein domain. To address the latter, binding preferences were established based on two parameters: (i) the mutation enrichment score, which is based on the NGS counts of all pairs of wild-type and mutant peptides for a given domain-mutation pairs (see methods) and (ii) the $p$ value of a Mann–Whitney test performed on those counts. Plotting the mutation enrichment score of each domain-mutation pair against the $p$ value of the Mann–Whitney test gives rise to a V-shaped plot. The left arm indicates interactions for which mutations diminish binding and the right arm interactions which are enhanced by mutations (Fig. 2D). We used a $p$ value $\leq 0.001$ of the Mann–Whitney test to define a set of domain-mutation pairs for which the mutation modulates the interaction significantly. Based on the chosen cut-off values, we found a total of 275 mutations to modulate 279 SLiM-based PPIs. This constitutes 366

domain-mutation pairs, of which 173 are diminished (or disrupted) and 193 are enhanced (or created) by the mutations (Fig. 2A,D). Some mutations have dual functions, disrupting interactions with one bait protein while creating a binding site for a second protein. Alternatively, mutations were found with the same effect on binding for several bait protein domains of the same family, exemplified by mutations in the PPxY motif that affect the binding of WW domains. Since the significance of the mutation modulation is calculated on collapsed peptide pairs, we further explored in how many cases there is a significant difference in binding for individual peptide pairs. We found for 201 out of the 366 mutation-modulated domain-mutation pairs at least one individual peptide pair indicating significant binding differences between wild-type and mutant peptide (Dataset EV6; Appendix Fig. S3). Finally, we note that relaxing the $p$ value cut-off to $p$ value $\leq 0.01$ would increase the numbers to 854 domain-mutation pairs affected by mutation with half being promoted (429) and half being diminished (425) by the mutation.

Interactions affected by mutations were found for about 70% of the bait proteins. With respect to the contribution of the different bait protein categories defined above (Fig. 1E), we found that baits categorised as cancer-associated proteins (Zheng et al, 2021; Cheng et al, 2021) enriched for a proportionally smaller set of perturbed interactions than expected from the number of interactions they engaged in (32% of the total number of interactions, but only 23% of the perturbed interactions) (Fig. 2B). For other bait categories the distribution of found versus perturbed interactions was fairly equal. In terms of previously reported interactions, we found an overlap of 38 unique interactions reported as onco-PPIs (Cheng et al, 2021) with our data, and the loss of binding upon mutations described in the onco-PPI dataset matched with our results for 18 mutations, with the majority of those representing mutations affecting the Kelch-like ECH-associated protein (KEAP1)-NFE2L2 interaction (Dataset EV7). For the set of proteins belonging to the category "PPIs disrupted by mutations" (Fragoza et al, 2019), we found only one interaction matching our selection results, which did not match in terms of probed mutations (Dataset EV7). We further observe for this bait domain category approximately equal numbers of interactions to be weakened or enhanced by mutation (50 weakened vs 62 enhanced), which is a general trend for most of the bait proteins. Exceptions to this trend include the WW domains, the KEAP1 KELCH domain and the importin subunit α-3 (KPNA4) ARM domain, for which we identified more mutation disrupted interactions, and the autophagy-related protein 8 (ATG8) proteins and peripheral plasma membrane protein CASK (CASK), for which we found more mutation-enhanced interactions (Fig. 2C).

## Validation of GenVar_HD2 selection results by affinity measurements

We selected 24 wild-type/mutant peptide pairs binding to 15 distinct bait proteins (1–4 peptide pairs per domain) and validated the effect of the disease-associated mutations by affinity measurements (Dataset EV8). Affinities of the wild-type and mutant peptides were determined ($K_D$-values) using a fluorescence polarisation (FP) displacement assay (Fig. 2E; Table 1; Appendix Figs. S4,5; Dataset EV9). The assay is based on the

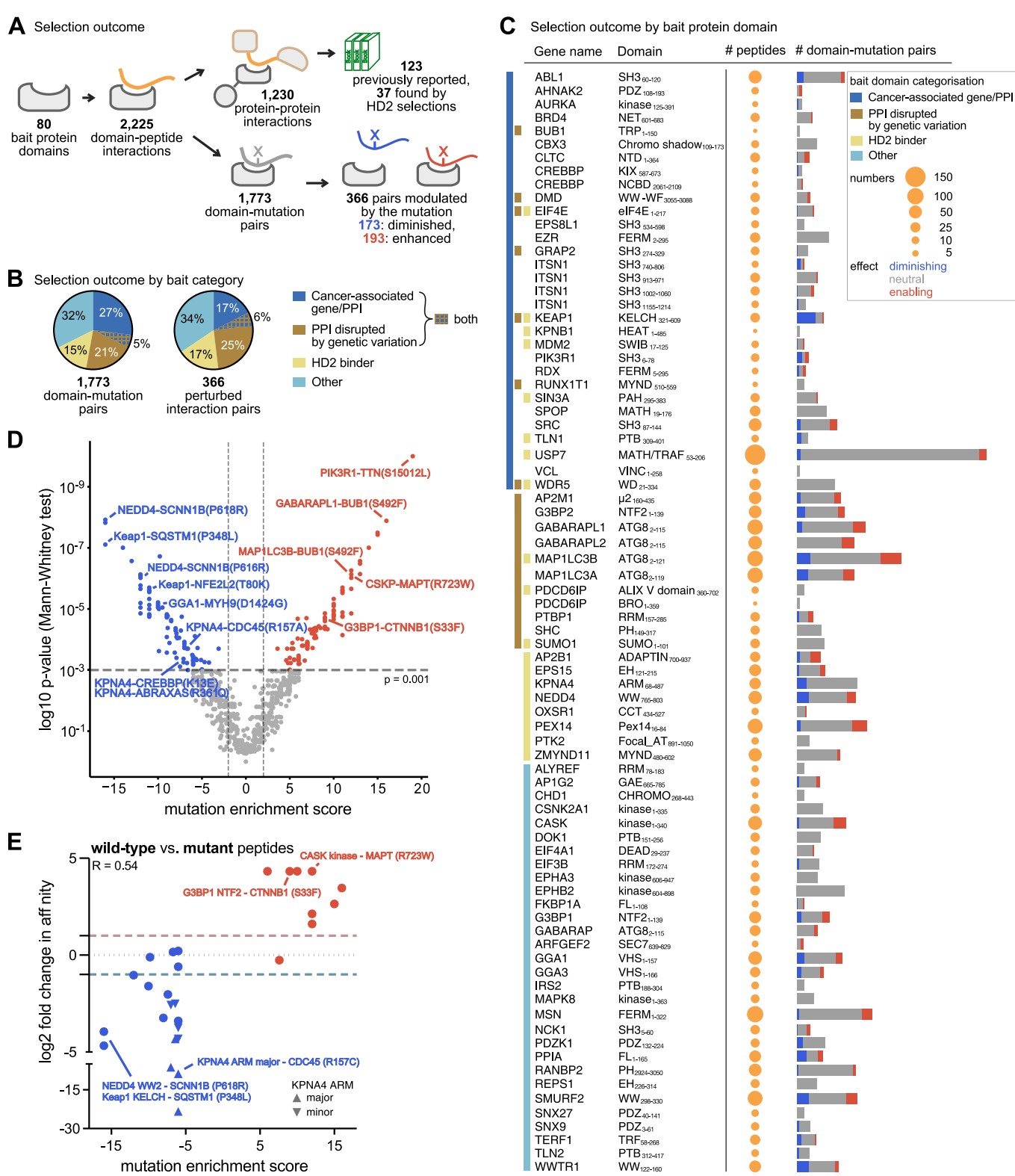

**A** Selection outcome

**B** Selection outcome by bait category

**C** Selection outcome by bait protein domain

**D**

**E** wild-type vs. mutant peptides

complex formation of a fluorescein isothiocyanate (FITC)-labelled peptide with its respective bait protein domain and displacing the FITC-labelled peptide sequentially by titrating unlabelled wild-type or mutant peptide. We validated high/medium-confidence

interactions, and further included a low-scoring KPNA4-binding ZNF526$_{152-167}$ peptide, since we judged the K160T mutation in the putative NLS likely to abrogate binding. Moreover, we found a BCL11A$_{44-56}$ peptide as a ligand for both the clathrin N-terminal

◄   **Figure 2.   Overview of the GenVar_HD2 phage display selections and validations.**

(A) Schematic overview of the phage selection outcome. (B) Summary of selection results by bait categories. (C) ProP-PD selection outcome by bait protein domain. The colour of the bar on the left indicates the bait category. The circle size encodes the number of found binding peptides and the bar on the right indicates the number of domain-mutation pairs (red: mutation enhances, blue: mutation diminishes and grey: neutral effect of the mutation). (D) V-shaped plot depicting the domain-mutation pairs plotted with their mutation enrichment score against the $p$ value of the Mann–Whitney test. Interactions, which are significantly modulated by the mutation (Mann–Whitney test: $p$ value ≤0.001), are coloured in red to indicate an enabling effect and in blue to indicate a disabling effect of the mutation. (E) Correlation of the log2 fold-change in affinity between wild-type and mutant peptides with the mutation enrichment score (Spearman $r = 0.54$). A positive correlation is judged by at least a twofold change in affinity and indicated by the dotted lines. If within a peptide pair, one of the peptides (wild-type or mutant) did not bind (no displacement), the fold-change was set for visualisation to 20 (log2 value: −4.3 or 4.3) (Table 1). Source data are available online for this figure.

domains (CLTC NTD) and G3BP2 NTF2 domain, with the C47F BCL11A mutation enabling the binding in both cases (Table 1), and consequently determined the affinities for both domains. The measured affinities ranged from nM to high µM in agreement with previous reports on affinities of motif-based PPIs (Table 1). We cross-validated the fold-change in affinity for the binding of the MAP1LC3B ATG8 domain to wild-type/mutant (T298I/V299L/V300R) BRCA2$_{292-307}$ peptide by isothermal titration calorimetry (Appendix Fig. S6; Dataset EV10), which confirmed the higher affinity of the domain for the mutant ($K_D = 2$ µM) over the wild-type peptide ($K_D \approx 20$–100 µM). For the KPNA4 ARM domain, we determined affinities for both the minor and major pocket. In all cases, we found the peptides bind with higher affinity to the major pocket (Table 1), which is expected as the tested peptides contain a classical KRx(R/K)-binding motif that is recognised by this pocket.

The affinity determinations revealed that the binding effect of 19 mutations (79%) was in agreement with the GenVar_HD2 selection results in terms of higher affinity for either wild-type or mutant peptides and as judged by at least twofold difference in $K_D$-values (Fig. 2E; Table 1). This compares well with the previous results of the phosphomimetic selection screen, which found 82% of the sampled cases in good agreement with the mutagenic phage selection results (Kliche et al, 2023). We hypothesised that contextualising the mutations by mapping them to the SLiM-consensus of the respective binding domains could serve as additional filtering criterion for hit prioritisation. This is based on the assumption that mutations of key residues within a SLiM are more likely to affect binding. We categorised 15 mutations of the 24 domain-mutation pairs as motif mutations. Of those, 13 mutations were found to cause significant changes in affinity, which increased the agreement between the GenVar_HD2 selection results and the affinity measurements to 87%. Thus, the results support the expectation that mutations affecting key residues of the motif will have a stronger effect on binding. However, even though mutations occurring within core positions of motifs are more likely to result in the reduction or enhancement of affinity, there is a growing literature reporting on the contribution of non-consensus residues in motif instances (wild-card positions) and of motif-flanking regions to the affinity of interactions (Karlsson et al, 2022; Bugge et al, 2020). Here, we note, for example, that the affinity of the ADP-ribosylation factor-binding protein GGA1 (GGA1) VHS domain to the myosin-9 peptide (MYH9$_{1419-1433}$), is diminished by a D1424H mutation of the first wild-card position of the D**D**LLV sequence (motif DxxLV; Table 1). Thus, while limiting the analysis to key positions increases the precision of the analysis it comes at

the cost of excluding motif-flanking or wild-card mutations that may tune the affinities of interactions.

## Prioritisation of mutations by motif consensus mapping

To provide a high confidence set of interactions that are enabled/disabled by disease mutations, we mapped the motif consensus of the bait protein domains (Dataset EV2) to the wild-type/mutant peptide stretches in the prey protein (Dataset EV6). For the 366 mutation-modulated pairs, we found for 298 pairs that the binding region contained the motif consensus of the respective bait protein domain either in the wild-type or mutated sequences (Fig. 3A). We considered the distance of the mutation in relation to the motif consensus since this is the main determinant for predicting the impact of the mutation on binding. For this, we categorised the mutations, based on their position relative to the motif consensus, as key residues, wild-card positions and flanking positions. We found, for the 298 mutation-modulated interactions with a mapped motif consensus, that 110 mutations affect key residues, of which 75 perturbed existing SLiMs and 35 mutations created novel motif instances (Fig. 3A; Dataset EV6). In addition, 85 mutations map to wild-card positions of existing motifs and 103 mutations were found in the motif-flanking regions. Among those, we consider the mutations in key residues and those creating a motif consensus to define our high confidence dataset.

As can be expected, almost all motif-creating mutations are enabling interactions and most of the mutations affecting key residues in existing motifs are negatively affecting the interactions in our phage display (Fig. 3B). Mutations of wild-card or flanking residues both enhance and diminish binding, pinpointing their ability to tune interactions. The effect of the mutations on binding may also depend on how conservative the mutation is, so that substitutions between hydrophobic residues might evoke minor affinity difference even if occurring within the key residues. To assess this systematically, we calculated the Grantham score (Grantham, 1974) for the different mutations, which assesses the chemical similarity between amino acids and categorises the mutations into conservative, moderately conservative, moderately radical and radical mutations. We found that the motif-creating mutations have a higher fraction of (moderately) radical mutations and that mutations affecting key residues of existing motifs have a higher proportion of moderately conservative mutations (Fig. 3C).

Together, the analysis consolidates that motif and mutation mapping can be used for hit prioritisation, and that the set of interaction-enhancing mutations is enriched for cases in which

**Table 1.   Overview of affinity data.**

| Protein domain | Peptide sequence | Gene name and mutation | $K_D$-values [µM] + SEM | | Mutation enrichment score; p value |
|---|---|---|---|---|---|
| | | | wild-type | mutant | |
| ABL1 SH3 | $_{64}$-YPRMPEAA(P/R)PVAPAPAAP$_{-80}$ | TP53 P72R | 650 ± 30 | 560 ± 30 | −6; 6.30E-04 |
| CASK kinase domain | $_{717}$-YSGDTSP(R/W)HLSNVSST$_{-731}$ | MAPT R723W | n.b. | 39 ± 4 | 12; 9.60E-07 |
| | $_1$-MSD(S/L)WVPNSASGQDPG$_{-16}$ | FANCA S4L | n.b. | 15.0 ± 0.7 | 6; 6.30E-04 |
| CLTC NTD | $_{44}$-DLLT(C/F)GQAQMNFP$_{-56}$Y | BCL11A C48F | 1100 ± 380 | 361 ± 43 | 12; 9.60E-07 |
| | $_{258}$-SGNLIDL(Y/C)GNQGLP$_{-271}$Y | MITF Y265C | 191 ± 27 | 206 ± 13 | −9.8; 1.90E-07 |
| CREBBP KIX | $_{2847}$-SDIMDFV(L/Q)KNTPSMQ$_{-2861}$Y | KMT2A L2854Q | 380 ± 30 | 340 ± 10 | −6,7; 2.00E-04 |
| GABARAPL1 ATG8 | $_{486}$-DKDEWQ(S/F)LDQNEDAFE$_{-501}$ | BUB1 S492F | 5.5 ± 0.7 | <0.3 | 16; 1.30E-08 |
| G3BP1 NTF2 | $_{1793}$-EFPV(L/P)FFGSNDYLWTH$_{-1808}$ | NSD1 L1797P | 12 ± 2 | 49 ± 1 | −7.4; 5.10E-05 |
| | $_{27}$-QQSYLD(S/F)GIHSGATTT$_{-42}$ | CTNNB1 S33F | n.b. | 57 ± 2 | 9; 1.90E-05 |
| G3BP2 NTF2 | $_{44}$-DLLT(C/F)GQAQMNFP$_{-56}$Y | BCL11A C48F | n.b. | 215 ± 43 | 10; 6.90E-06 |
| GGA1 VHS | Y$_{1419}$-QQELD(D/H)LLVDLDHQR$_{-1433}$ | MYH9 D1424H | 101 ± 3 | 306 ± 3 | −10; 8.00E-06 |
| KEAP1 KELCH | $_{343}$-SKEVD(P/L)STGELQSL$_{-356}$Y | SQSTM1 P348L | 0.9 ± 0.1 | 23 ± 3 | −16; 7.80E-08 |
| KPNA4 ARM major | $_{349}$-LDLDDRWQFKRS(R/Q)LLD$_{-364}$ | ABRAXAS1 R361Q | 0.17 ± 0.02 | 75 ± 3 | −6; 4.50E-04 |
| | Y$_5$-LLDGPPNP(K/E)RAKLSS$_{-19}$ | CREBBP K13E | <0.00008 | 860 ± 20 | −6; 4.50E-04 |
| | $_{152}$-EPSEK(R/C)TRLEEEIVE$_{-166}$Y | CDC45 R157C | 0.090 ± 0.003 | 7 ± 1 | −7; 1.70E-04 |
| | $_{152}$-PELWVAHR(K/T)AQHLSAT$_{-167}$ | ZNF526 L160T | 135 ± 3 | n.b. | −6.4; 4.80E-04 |
| KPNA4 ARM minor | $_{349}$-LDLDDRWQFKRS(R/Q)LLD$_{-364}$ | ABRAXAS1 R361Q | 22 ± 3 | 297 ± 16 | −6; 4.50E-04 |
| | Y$_5$-LLDGPPNP(K/E)RAKLSS$_{-19}$ | CREBBP K13E | 153 ± 5 | n.b. | −6; 4.50E-04 |
| | $_{152}$-EPSEK(R/C)TRLEEEIVE$_{-166}$Y | CDC45 R157C | 91 ± 11 | 545 ± 70 | −7; 1.70E-04 |
| | $_{152}$-PELWVAHR(K/T)AQHLSAT$_{-167}$ | ZNF526 L160T | 69 ± 1 | 395 ± 10 | −6.4; 4.80E-04 |
| KPNA7 ARM major | $_{152}$-EPSEK(R/C)TRLEEEIVE$_{-166}$Y | CDC45 R157C | 6.6 ± 0.7 | 380 ± 15 | n.d. |
| MAP1- LC3A ATG8 | $_{209}$-QTKWE(L/P)LQQVDTSTRT$_{-224}$ | KRT1 L214P | 55 ± 5 | 580 ± 30 | −6; 3.90E-04 |
| MAP1- LC3B ATG8 | $_{486}$-DKDEWQ(S/F)LDQNEDAFE$_{-501}$ | BUB1S492F | 7 ± 1 | 1.6 ± 0.1 | 12; 7.40E-07 |
| | $_{334}$-GDDDWTHLSSKEVD(P/L)S$_{-349}$ | SQSTM1 P348L | 0.5 ± 0.1 | 0.6 ± 0.1 | 7.6; 1.90E-04 |
| | $_{292}$-EDEVYE(T/I)(V/L)(V/R)DTSEEDS$_{-307}$ | BRCA2 T298I/V299L/ V300R | 3.12 ± 0.08 | 0.52 ± 0.01 | 15; 3.70E-08 |
| NEDD4 WW2 | $_{611}$-PIPGT(P/R)PPNYDSLRLQ$_{-626}$ | SCNN1B P616R | 39 ± 3 | 80 ± 7 | −12; 7.40E-07 |
| | $_{611}$-PIPGTPP(P/R)NYDSLRLQ$_{-626}$ | SCNN1B P618R | 39 ± 3 | 603 ± 73 | −16; 1.20E-08 |

**Table 1.** (continued)

| Protein domain | Peptide sequence | Gene name and mutation | $K_D$-values [µM] + SEM | | Mutation enrichment score; p value |
|---|---|---|---|---|---|
| | | | wild-type | mutant | |
| PPIA | $_{2993}$-GASSPSYG(P/S)PNLGFVD$_{-3308}$ | KMT2D P2301S | 177 ± 20 | 1675 ± 144 | −8; 5.10E-05 |
| USP7 MATH/ TRAF | Y$_{38}$-ISSSSTSTMPNSS(Q/K)SS$_{-53}$ | CHK2 Q51K | 3.1 ± 0.2 | 4.7 ± 0.3 | −6; 3.60E-04 |

The protein domain used, the peptide sequence, the mutation site sampled, the obtained $K_D$-values with SEM, as well as the associated mutation enrichment score and p value are indicated. The FP-monitored affinity measurements were performed in technical triplets. (n.b. not binding within concentration range used; n.d. not determined).

moderately radical mutations are occurring in key residues of the motifs.

## Exploring the PPI data in the context of the enabling and disabling mutations

We next explored the dataset in terms of which interactions are lost and gained by mutations. By analysing the number of interactions found for the different types of bait proteins we first notice some variation such that interactions with bait proteins involved in the ubiquitin system (E3 ligases) and transcriptional regulation are more commonly lost than enabled upon mutations. In contrast, interactions with proteins involved in autophagy (e.g. ATG8 proteins) or scaffolding are more commonly enabled than disabled by mutations (Fig. 4A; Dataset EV2C, EV6), as discussed further below. To investigate how previously reported PPIs are affected by mutations we mapped them on the V-shaped plot and assessed their distribution on the interaction-disabling and -enabling arms. Of the 366 interactions perturbed by mutations, 64 mutations affected 49 previously reported PPIs, since some mutations mapped to the same interacting prey protein. Of these mutations, 46 were found to reduce (or abrogate) interactions while 19 enhanced (or enabled) interactions (Fig. 4B).

Of the domain-mutation pairs found to be disrupted by mutations, 29% affect previously known interactions, which to some extent allows the cellular effects of these mutations to be inferred. For example, we identified several mutations (P616R, P616S, P617H, P618R and Y620H) overlapping the PPxY-motif of the Amiloride-sensitive sodium channel subunit beta (SCNN1B) that modulate binding to the E3 ubiquitin-protein ligase NEDD4 (NEDD4) WW2 domain (Fig. 4C). Through affinity determinations we confirmed weaker binding of the NEDD4 WW2 domain to the SCNN1B$_{611-626}$ peptide as a consequence of the P618R mutation at the p + 2 position of the PPxY-binding motif (39 µM vs ≈600 µM), and a minor loss of affinity by the P616R mutation at the p-1 position (39 µM vs 80 µM) (Fig. 4C; Table 1). The loss of interaction with the E3 ligase found through the phage selections would suggest an altered SCNN1B protein level as a consequence of the mutations. Indeed, the turnover and cell surface expression of SCNN1B is reportedly regulated by NEDD4 and/or NEDD4-like (NEDD4L) (Rotin and Staub, 2012; Snyder et al, 2004; Staub et al, 2000, 1996). Mutations of the PPxY-motif in SCNN1B are linked to Liddle syndrome 1, a hereditary form of hypertension caused by augmented Na$^{2+}$ transport, and result in the loss of interaction with the HECT type E3 ligases and increased stability of the transmembrane protein (Enslow et al, 2019). Similarly, our analysis pinpointed several mutations of the nuclear factor erythroid-derived 2-like 2 (NFE2L2) disrupting its interactions with the E3 ligase KEAP1. The KEAP1-NFE2L2 pathway controls key aspects of the cellular redox balance, with NFE2L2 acting as a transcriptional activator. Loss of the interaction on account of mutation of the KEAP1-binding TGE motif in NFE2L2 leads to increased abundance of NFE2L2 and results in the resistance of cancer cells to reactive oxygen species (Taguchi and Yamamoto, 2017). The mutational ProP-PD thus efficiently identifies loss-of-function mutations with functional consequences.

Only 14% of the interactions enhanced by mutations were previously reported (27 domain-mutation pairs), these mutations may either enhance the affinity of an existing SLiM for its binding domain or may create additional interaction interfaces between the two proteins. For example, we found a cancer-related D695Y mutation in the tumour suppressor Breast cancer type 1 susceptibility protein (BRCA1) that creates novel binding sites for the Yx[FILV]-binding FERM domains of Moesin (MSN) and Radixin (RDX) (Ali et al, 2023). BRCA1 has previously been reported to interact with MSN/RDX through its BRCT domain, an interaction that localises the protein to the leading edges and focal adhesion sites in breast cancer cells. The novel FERM-binding motif in BRCA1 might serve to stabilise this interaction. However, most of the peptide-domain interactions that were created or enhanced by mutations are novel PPIs, that is neo-interactions. Many of the neo-interactions (21.3%) involve the autophagy-related ATG8 proteins (Figs. 2C, 4A,D). For example, we found a S492F mutation in the mitotic checkpoint serine/threonine-protein kinase BUB1 (BUB1), which was confirmed by affinity measurements to confer a four-fold increase in affinity for MAP1LC3B ATG8 (7 µM vs 1.6 µM; Table 1; Fig. 4D). We also found several mutations leading to neo-interactions with protein domains involved in scaffolding and trafficking, which may lead to aberrant localisation of the mutant proteins. One such example is the R723W mutation in the microtubule-associated protein tau (MAPT), associated with frontotemporal dementia/Parkinson's disease, that creates a novel binding site (p + 1 position of the [ILPV]W-motif) for the kinase domain of the peripheral plasma membrane protein CASK (CASK) (no binding vs 39 µM; Fig. 4E; Table 1). We further uncovered a cancer-associated S33F CTNNB1 mutation which generates a novel binding site for G3BP1/2 (p + 3 position of the [FILV]xFG-motif). Through affinity measurements, we confirmed that the S33F mutation turned the CTNNB1$_{27-42}$ peptide from a non-binder of G3BP1 NTF2 to a ligand with micromolar affinity (no binding vs 57 µM; Fig. 4F; Table 1). The S33F substitution may thus have a dual function in creating a binding site for G3BP1 and simultaneously destroying the CTNNB1 degron that binds to β-TrCP (Mészáros et al, 2017).

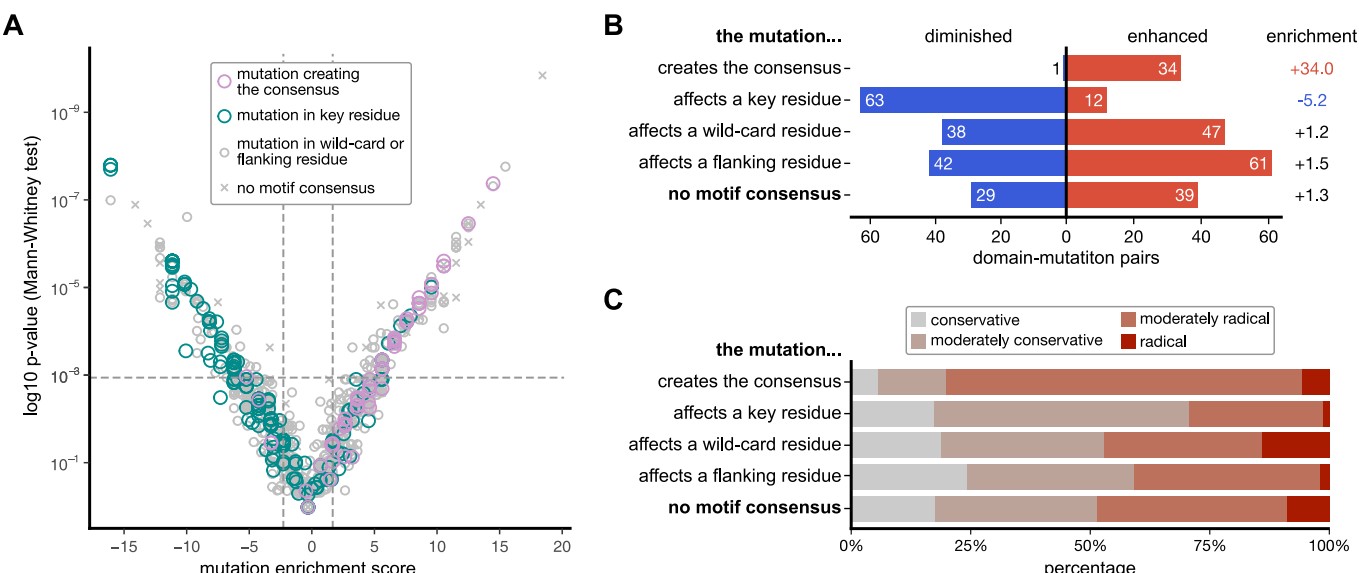

**Figure 3. Motif instance mapping and evaluation of the impact of mutations in relation to the motif.**

(A) V-shaped plot depicting the domain-mutation pairs plotted with their mutation enrichment score against the *p* value of the Mann–Whitney test. Domain-mutation pairs where the mutation is within the key residues of the motif instance are in green and those where mutations create the motif instance are in purple. In grey are the pairs, for which the mutation is found in wild-card or flanking residues, and pairs for which the motif of the bait protein domain was not found (x). (B) Overview of the effects of the mutations depending on their position in relation to the key binding determinants. (C) Motif categories of the domain-mutation pairs with the categorisation of the mutation according to the Grantham score (conservative (<51), moderately conservative (51–100), moderately radical (101–150) or radical (>150)).

Of note, an association between CTNNB1 and G3BP1 has previously been suggested based on a SILAC-based proteomics experiment (Rosenbluh et al, 2016).

In summary, through selections with the GenVar_HD2 library against 80 bait protein domains we identified 366 domain-mutation pairs that are disrupted, diminished, reinforced or created by the associated mutations. The results provide novel insight into how PPI networks are perturbed and rewired by mutations.

## Validating the binding-disabling effect of mutations in the context of full-length proteins

We next aimed to validate interactions and the impact of mutations in the context of the full-length proteins by co-immunoprecipitation. Three of the interactions tested were enabled by the mutations and validated on the peptide level (binding of G3BP1 to S33F CTNNB1, CASK to R723W MAPT (MAPT-D isoform: R348W), and MAP1LC3B to S492F BUB1). However, we were unable to confirm the effect of the enabling mutations in the context of the full-length proteins in HeLa (and/or HEK293T) cells (Appendix Fig. S7A). Thus, while we validated the neo-interactions through biophysical affinity measurements, the interactions might not be strong enough to compete with the pool of endogenous interactors that the prey protein encounters in a cellular setting.

Three interactions tested were weakened by the mutation, one being a known interaction between KEAP1 and SQSTM1 (Komatsu et al, 2010), which we found to be disrupted by a P348L mutation in SQSTM1. We found the SQSTM1$_{343-356}$ peptide to bind with 0.9 μM to KEAP1 KELCH and that the P348L mutation confers a 25-fold loss of affinity (Fig. 5A,B; Table 1), which is also reflected in

a loss of interaction between the full-length proteins (Fig. 5C). The SQSTM1 P348L mutation is linked to frontotemporal dementia and amyotrophic lateral sclerosis 3 (Rubino et al, 2012; Goode et al, 2016). Our findings are in line with a previous study, which also showed that the mutation results in less NFE2L2-mediated transcription of anti-oxidative stress genes (Goode et al, 2016). Of note, we also assessed the impact of the P348L SQSTM1 mutation on the known interaction with MAP1LC3B, since the display indicated an enabling impact of the mutation on the interaction, which was also proposed by a previous report (Deng et al, 2020). We found, however, in our affinity measurements, that the mutation has no impact on binding (Appendix Fig. S5). While our co-immunoprecipitation experiment confirmed the interaction between MAP1LC3B and wild-type SQSTM1, the results were inconclusive with respect to the effect of the P348L SQSTM1 mutation, and the combined results would thus suggest that the mutation has little effect on MAP1LC3B binding (Appendix Fig. S7B).

The other two interactions tested that were weakened by mutations involved BRCA1-A complex subunit Abraxas 1 (R361Q ABRAXAS1) and CDC45 (R157C CDC45). Both proteins bound to the KPNA4 ARM domain through putative NLSs KRx(R/K)-motif; Table 1; Fig. 5D). ABRAXAS1 is implicated in DNA repair and exerts tumour suppressive functions through its interactions with BRCA1 (Castillo et al, 2014; Solyom et al, 2012; Bose et al, 2019). The R361Q ABRAXAS1 mutation is associated with breast cancer due to increased genome destabilisation (Bose et al, 2019; Solyom et al, 2012). CDC45 is part of the helicase complex responsible for DNA unwinding during DNA replication and CDC45 mutations, including the R157C mutation, have been associated with Meier-Gorlin syndrome, a developmental condition characterised by short

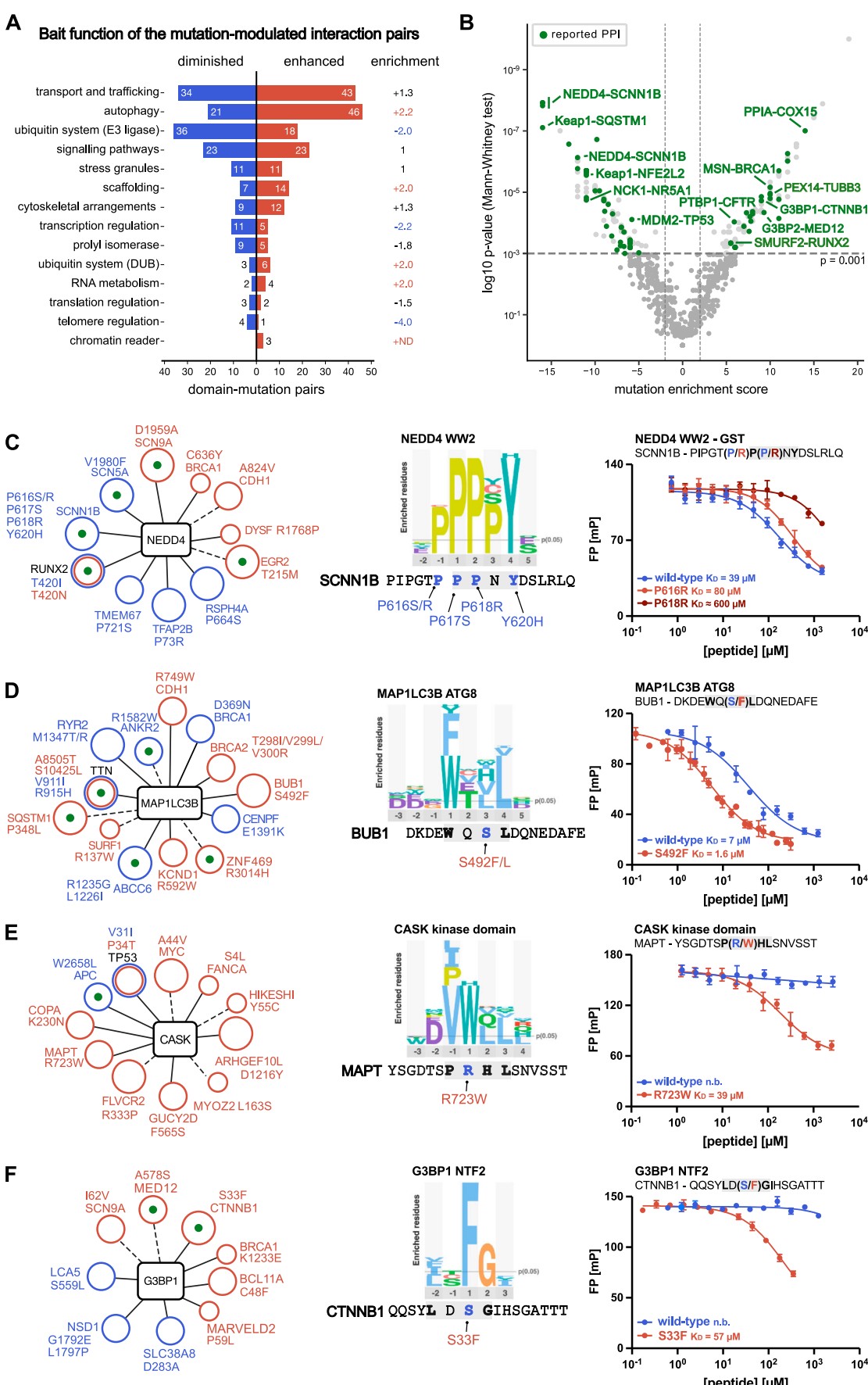

◄ **Figure 4. Exploring the PPI data underlying the mutations.**

(A) Domain-mutation pairs enhanced or diminished by mutations as categorised based on the functional processes in which the bait protein domains are involved. (B) V-shaped plot depicting the domain-mutation pairs plotted against the mutation enrichment score and with mapped previously reported interactions (in green). A select set of pairs are indicated with names. (C–F) Left: Network of NEDD4 (C), MAP1LC3B (D), CASK (E) and G3BP1 (F) with found mutation-modulated interactions indicated (blue: weakened interaction; red: enhanced interaction). A green dot indicates a previously reported interactor. A dotted line indicates that the expected consensus motif for the bait is missing in the found peptide, and a full line indicates that the motif is present in the peptide. The circle size encodes the confidence level of the found domain-mutation pair. (C–F) Middle: Consensus motifs as established by previous selections against the HD2 library. (C–F) Right: Fluorescence polarisation-monitored displacement experiments measuring the affinities of NEDD4 WW2 with wild-type and mutant (P616R and P618R) SCNN1B$_{611-626}$ peptides, MAP1LC3B ATG8 domain with wild-type and mutant (S492F) BUB1$_{486-501}$ peptides, CASK kinase domain with wild-type and mutant (R723W) MAPT$_{717-731}$ peptides, and G3BP1 NTF2 domain with wild-type and mutant (S33F) CTNNB1$_{27-42}$ peptides. Measurements were in technical triplets and displayed is the mean with standard deviation. Source data are available online for this figure.

stature (Fenwick et al, 2016). We confirmed by affinity measurements that the R361Q ABRAXAS1 mutation (position p + 4 of the motif) and the R157C CDC45 (position p + 2 of the motif) weakened the binding to both the major and minor pocket of the KPNA4 ARM domain (major pocket: ABRAXAS1: 400-fold; CDC45: 70-fold; Fig. 5E; Table 1). We tried to confirm the effect of the mutations on the interaction of ABRAXAS1 and CDC45 with KPNA4 through co-immunoprecipitation, but with negative results (Appendix Fig. S7C). As there are seven importin-α variants in the cell but only KPNA4 was included as bait in the phage selection, we explored if we could validate interactions with other importin-α proteins (KPNA1, -2, -3, -6 and -7) through co-immunoprecipitation since the importin family members bind similar motifs (Kosugi et al, 2009). We thereby established an interaction between KPNA7 and wild-type CDC45, which was abrogated by the R157C CDC45 mutation (Fig. 5F). We also confirmed by affinity measurements that the R157C mutation abolishes the binding of the CDC45$_{152-166}$ peptide to the major pocket of KPNA7 (50-fold; Table 1; Fig. 5G). Interestingly, KPNA7 has been suggested to mainly play a role during embryogenesis and to be associated with neurodevelopmental diseases (Wang et al, 2012; Oostdyk et al, 2020; Paciorkowski et al, 2014), which might hint at a shared physiological context with CDC45 and an importance of their interaction in development. We further wondered if the mutations of the putative NLSs would perturb the cellular localisation of the proteins, as that would be the expected functional consequence and has been suggested for the R361Q ABRAXAS1 mutation (Solyom et al, 2012; Bose et al, 2019). For this, we transiently expressed EGFP-tagged wild-type/R361Q ABRAXAS1 or wild-type/R157C CDC45 in HEK293T cells and assessed whether the mutations conferred a change in the EGFP localisation (Fig. 5H,I; Appendix Fig. S8–11). Both wild-type ABRAXAS1 and wild-type CDC45 were found to co-localise with staining for the cell nuclei, in line with the nuclear localisation of both proteins (Pollok et al, 2007; Solyom et al, 2012). In contrast, the R361Q ABRAXAS1 and R157C CDC45 mutants showed a more cytoplasmic localisation (Fig. 5H,I, Appendix Figs. S8–11). Thus, the GenVar_HD2 screen identified functional NLSs in ABRAXAS1 and CDC45 that are abrogated upon mutations.

Overall, we demonstrate that the mutational ProP-PD selections successfully identified mutations disrupting SLiM-based interactions, of which a subset were confirmed on the level of the full-length proteins. We further pinpoint the functional consequences of the R361Q ABRAXAS1 and R157C CDC45 mutations and uncover an interaction between CDC45 and KPNA7.

## Mapping back to the diseases underlying the mutations

As a high level analysis, we investigated globally which diseases are linked to mutations with enabling or disabling effects on binding. We used clustered disease categories (Dataset EV1C,D) and mapped them to the domain-mutation pairs (Dataset EV6). We observed a similar distribution of the disease categories among the perturbed domain-mutation pairs as in the GenVar_HD2 library design (Fig. 1D), with the main clusters represented by the mixed category (85), neurological (42) and musculoskeletal (28) diseases, as well as cancer (46) (Fig. 6A). The peptides with mutations associated with cardiovascular/haematologic diseases (535) disappeared after selection, possibly because there was no domain binding to these peptides as bait in our selection. Thus, no disease category stood out as being overrepresented in terms of being affected by mutations targeting SLiMs. In addition, the majority of the mutations modulating interactions are somatic mutations (82%), consistent with the distribution of somatic and germline mutations in the library design. Below, we highlight a select set of cases where the diseases are either associated with mutations in ion channels, or associated with cancer.

### Diseases associated with mutations in ion channels

Many of the interaction perturbing mutations linked to neurological, metabolic and cardiovascular/haematological diseases are found in ion channels (calcium channel: CACNA1H; sodium channels: SCN1A, -5A, -9A and SCNN1B; potassium channel: KCND1, -J2; non-selective cation channel: TRPC6) and affect interactions with HECT type E3 ligases or with autophagy-related ATG8 proteins (Fig. 6B). The altered interactions could affect channel activity or surface expression, as described above for the SCNN1B mutations (Fig. 4C). In addition, we found that a D1959A mutation in SCN9A, located at the p + 5 position of the PPxY-motif, enhanced the binding to several WW domains (e.g. NEDD4 WW2 and SMURF2 WW3 domains). The mutation is associated with epilepsy and neuropathy, and of note, downregulation of the E3 ligase NEDD4-2 is proposed to dysregulate SCN9A in neuropathic pain (Laedermann et al, 2013). A Y1958C SCN9A mutation, which maps to the tyrosine of the PPxY-motif and abrogates binding, has further been described to cause epilepsy (Zhang et al, 2020). Related to potentially perturbed receptor trafficking, we found an E897K TRPC6 mutation that confers a loss of binding to the μ2 subunit of the clathrin-AP2 adaptor (AP2M1). The mutation, associated with glomerulosclerosis (metabolic disease) and enhanced channel activity (Dryer and Reiser, 2010),

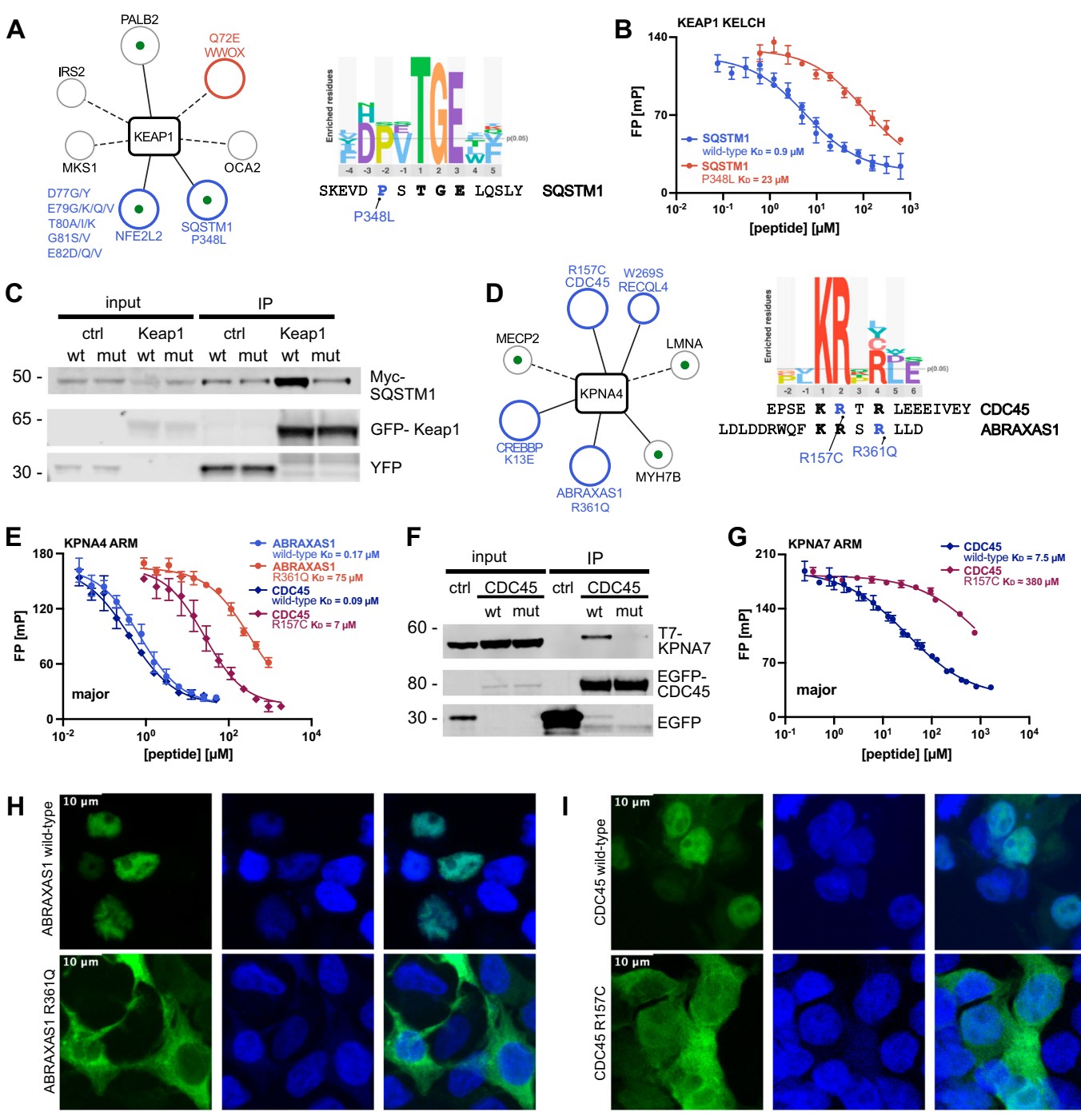

is found at the p + 2 position of the YxxΦ-motif of AP2M1. We hypothesise that the mutation results in impaired endocytosis and channel turnover.

## Cancer

Of the 62 domain-mutation pairs where the mutation is related to cancer (cancer category and cancer-related mutations in the mixed category), we found 28 to enable interactions and 34 to disable interactions. We investigated if the discovered cancer-related mutations represent hotspots in the TCGA data, so that the

mutation has been identified in more than one study. Among the cancer-associated proteins we found five hotspot mutations conferring loss (NFE2L2 E79V/Q) or gain (PMS2 R107W, MUTYH S321L, CTNNB1 S33F) of interactions (Fig. 6C; Dataset EV6).

Many of the cancer-related mutations are associated with breast and/or ovarian cancer, and map to BRCA1, BRCA2 and their associated proteins involved in processes related to DNA synthesis and repair (e.g. CHEK2, BARD1, BRIP1 and TP53). This is expected given the central role of these proteins and pathways in

**Figure 5.  Assessment of disruptive mutations on the cellular level.**

(A) Left: Network of KEAP1 and its found prey proteins. Interactions and corresponding mutations disrupted by the mutation are indicated in blue, enhanced in red and neutral effects in grey. A green dot indicates previously reported interacting prey proteins and a straight line if the domain's binding motif is found in the binding peptide. The circle size encodes the confidence level of the domain-mutation pairs. Right: Consensus motif of KEAP1 KELCH as established with the HD2 library and the aligned SQSTM1 peptide with the P348L mutation site indicated in blue. (B) FP-monitored displacement curves for KEAP1 KELCH domain and the SQSTM1$_{343-356}$ wild-type and P348L mutant peptides. Measurements were in technical triplets and displayed is the mean with standard deviation. (C) GFP trap of GFP-KEAP1 and probing for co-immunoprecipitation of wild-type/P348L Myc-SQSTM1 in HeLa cells ($n = 3$). (D) Left: Partial network of KPNA4 and its found prey proteins. The network shows interactions with previous literature support and interactions associated with mutations having a significant effect on binding (Mann–Whitney test: $p$ value ≤0.001). Colouring and size as in (A). Right: Consensus motif of KPNA4 ARM as established with the HD2 library and the aligned CDC45 and ABRAXAS1 peptides with the mutation sites indicated in blue. (E) Displacement curves for KPNA4 ARM domain with wild-type/R157C CDC45$_{152-166}$ and wild-type/R361Q ABRAXAS1$_{349-364}$ peptides. Measurements were in technical duplicates and displayed is the mean with standard deviation. (F) GFP trap of EGFP ($n = 3$), wild-type EGFP-CDC45 ($n = 3$) or mutant R157C EGFP-CDC45 ($n = 2$) and probing for co-immunoprecipitation of T7-KPNA7 in HEK293T cells. (G) Displacement curves for KPNA7 ARM domain with wild-type/R157C CDC45$_{152-166}$, sampling the major pocket. Measurements were in technical triplets and displayed is the mean with standard deviation. (H, I) Representative images of the localisation of wild-type and mutant EGFP-tagged (R361Q) ABRAXAS1 and (R157C) CDC45 in relation to the nuclear Hoechst staining ($n = 3$ for all except ABRAXAS1 wild-type for which $n = 2$). The full images are provided in Appendix S8–S11. Source data are available online for this figure.

cancer, the long IDRs of BRCA1 and BRCA2 and the large number of disease-associated mutations found in these regions. Other interesting observations relate to mutations in the tumour suppressor cadherin-1 (CDH1). We found two CDH1 mutations (V832L and A324V) linked to hereditary diffuse gastric adenocarcinoma that confer binding to the WW domain of the HECT type E3 ligase SMURF2, and a R794W mutation that creates an ATG8 binding site. These neo-interactions may lead to reduced CDH1 protein levels through proteasomal degradation and/or autophagy.

Finally, we highlight a talin 1 and 2 (TLN1 and TLN2) PTB binding site in the proto-oncogene tyrosine-protein kinase receptor Ret (RET), which is disrupted by two mutations (L1061P, Y1062A) linked to multiple endocrine neoplasia type 2 and to different types of tumours. Notably, the region has previously been described as a phospho-tyrosine-dependent DOK1 and SHC PTB-RET-binding site (Shi et al, 2004), which is lost upon the Y1062A mutation. Our results suggest that TLN1 and TLN2 PTB domains may bind unphosphorylated RET and might be implicated in RET signalling.

## Target development level and cancer essentiality of bait and prey proteins

The identification of interactions that are enabled/disabled by disease mutations may lead to the identification of novel targets for inhibition or modulation. We, therefore, explored the target development level of the prey and bait proteins engaging in mutation-modulated interactions (Fig. 6D). The target development level indicates the current druggability based on the classification in Tclin (proteins with approved drugs), Tchem (proteins known to bind small molecules), Tbio (proteins with well-studied biology) and Tdark (understudied proteins) (Sheils et al, 2020). Most of the prey and bait proteins engaging in mutation-modulated interactions are categorised as Tbio (bait protein domain: 40; prey protein: 131). However, a subset is reportedly druggable, and some of the proteins already have approved drugs (bait protein domain: Tchem: 11 and Tclin: 5; prey protein: Tchem: 20 and Tclin: 20, Dataset EV1E, EV2C, EV6).

In order to explore the results further, we analysed if the bait and/or prey proteins are essential for the growth of cancer cells based on the information in the Cancer dependency (DepMap)

portal (Fig. 6E; Dataset EV1E, EV2C, EV6). The DepMap portal accumulates data from large-scale functional genomic screens, hereby uncovering genes required for cancer cell growth and survival (Tsherniak et al, 2017). It provides a Chronos score, informing on the fitness effect of the gene knockout (Dempster et al, 2021). We plotted the Chronos score from CRISPR experiments for the bait proteins versus the prey proteins of the domain-mutation pairs. A score below $-0.5$ implies that the knockout of that gene results in growth inhibition, whereas a score below $-1$ indicates that the gene is essential for cancer cell growth. Most of the proteins are encoded by cancer non-essential genes (Chronos score $>-0.5$), which partially relates to the fact that the mutations in the library are linked to other diseases than cancer. We further overlayed the target development level information of our prey/bait proteins to their cancer gene essentiality. We found that 16 of the 62 cancer-associated domain-mutation pairs are with proteins encoded by genes with Chronos scores less than $-0.5$ and of those, ten have druggable prey or bait proteins. Among those, we find the aforementioned TLN1 PTB-RET interaction disrupted by L1061P or Y1062A and several BRCA1 interactions (Fig. 6E). Overall, there is limited overlap between cancer dependencies, their druggability and the domain-mutation pairs we found to be modulated by mutation.

## Discussion

With the enormous amount of reported missense mutations, we are facing a challenge to establish functional relationships between the observed genetic variation and a functional effect on the protein level. Deregulation of protein interactions is one of the key consequences of missense mutations (Sahni et al, 2015; Fragoza et al, 2019; Cheng et al, 2021). The extent to which mutations in SLiM-based interfaces contribute to disease progression is still poorly understood, and has as far as we know, not been experimentally explored in a systematic fashion. In this study, we performed a large-scale screening for disease-associated mutations that create, enhance, diminish or break SLiM-based interactions. We developed the GenVar_HD2 library that combines non-synonymous SNVs (12,301 unique mutations) with the unstructured regions of the human proteome, and applied it to elucidate the impact of the mutations on protein binding. We find, in selections against our bait protein domain collection, 275 mutations that affect 279 motif-based PPIs (366 pairs), with approximately half of

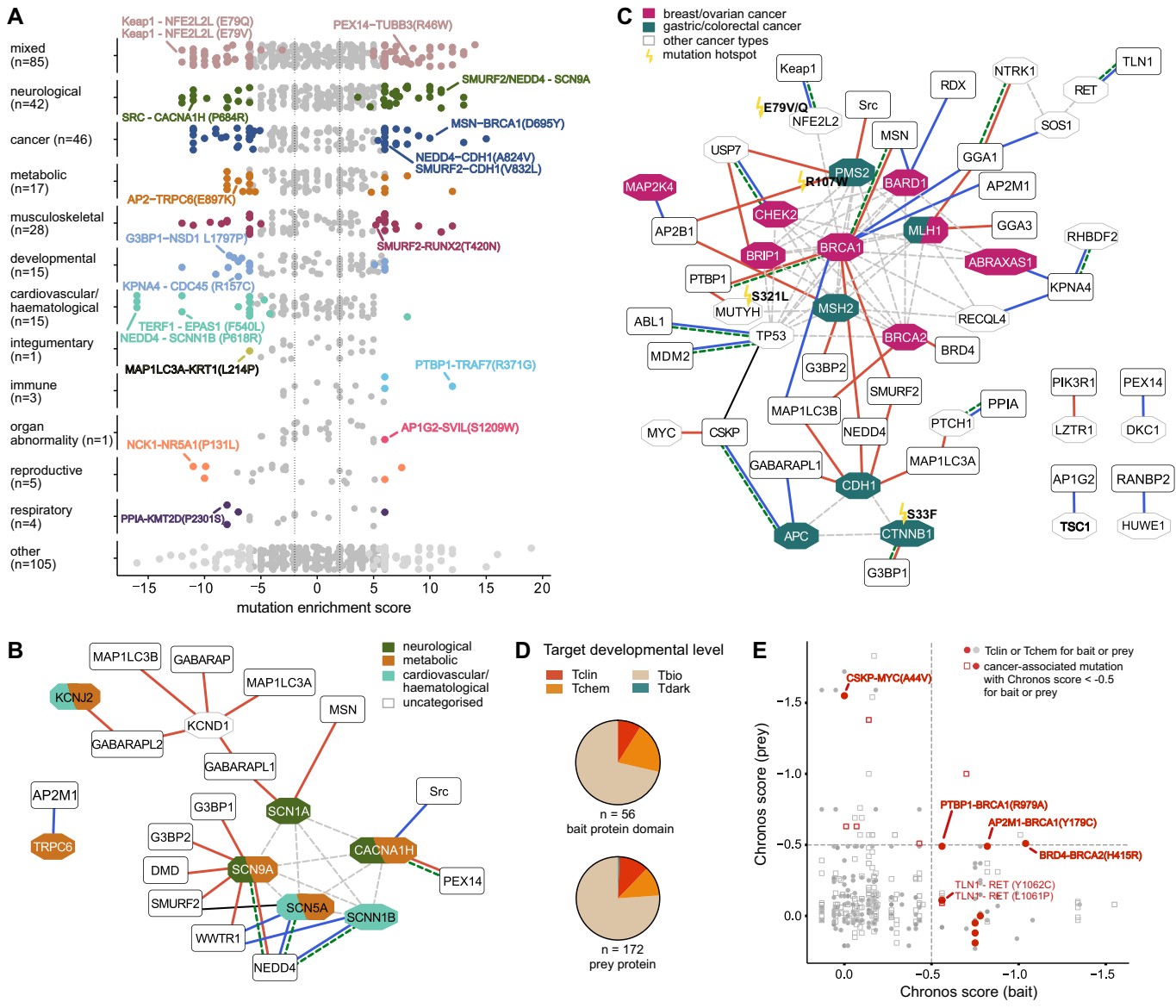

**Figure 6. Mapping back the mutations to the underlying diseases.**

(A) Domain-mutation pairs mapped back to the disease categorisation underlying the mutations. Colouring is by the *p* value of the Mann–Whitney test and the disease category (grey for *p* > 0.001, disease category colour for *p* ≤ 0.001) and directionality is by the mutation enrichment score (<0: diminished interaction, >0: enhanced interaction). (B) PPI network of interactions affected by mutations in ion channels (red lines: enhanced interactions, blue lines: diminished interactions; colouring of the prey proteins is by disease category of the associated mutations). Square shapes are for the bait proteins and octagonal shapes for prey proteins. Dotted edges indicate previously reported interactions between prey proteins and green dotted lines indicates when the bait–prey interaction has been previously reported in the literature. (C) PPI network of interactions affected by cancer-associated mutations (red lines: enhanced interactions, blue lines: diminished interactions). Square shapes are for the bait proteins and octagonal shapes for prey proteins. Dotted edges indicate previously reported interactions between prey proteins. A green dotted line indicates when the bait–prey interaction has been previously reported in the literature. Prey proteins with mutations associated with breast/ovarian or gastric/colorectal cancer are indicated in purple and turquoise, respectively. Mutations annotated as cancer hotspots in the TCGA data are indicated with a flash and detailed mutation information. (D) The target development level of prey and bait proteins engaging in mutation-modulated interactions. (E) CRISPR scores from the DepMap portal of the bait/prey protein-encoding genes. Filled circles indicate that the prey or bait protein has been categorised as Tclin or Tchem. Red indicates that the mutation is cancer-associated and that the CRISPR score of the gene encoding the prey or bait protein is <−0.5.

the mutations disrupting and half enhancing the interactions. Given that the recall of the phage-based interaction screening is about 20%, it can be estimated that we uncover a similar fraction of the interactions affected by mutations. Validation by biophysical affinity measurements showed that the results of the phage selections reflect binding preference to wild-type and mutant peptides accurately. Using motif

mapping, we further contextualised the mutations and found that mutations in key residues or mutations creating motifs commonly impact domain binding. Nonetheless, we also report on mutations in wild-card or flanking positions affecting binding. Moreover, while we could map the motifs for 298 domain-mutations pairs, we did not find the expected binding consensus for 68 of the pairs. This likely reflects

that some of the SLiM definitions used were overly strict, and may further indicate that some domains recognise variant motifs. Also, although applying strict filtering, we cannot exclude the presence of some false positive hits.

Most, if not all, mutant proteins for which we found perturbed interactions have more than one interactor based on the information provided in public databases (Del Toro et al, 2022, Szklarczyk et al, 2023). The missense SNVs that we found to impact PPIs target specific SLIMs and they will thus typically have edgetic effects, perturbing some but not all of the interactions of the mutant protein. In some cases, a given SLiM can bind to many members of a domain family, such as the PPxY-motif that binds to several WW domain-containing proteins. In such cases, a mutation will affect binding to all members of the family. In other cases, SLiMs are densely located in the IDRs and sometimes even overlap. In such cases, there are several potential outcomes of the mutations. The effect may be to abrogate several distinct interactions. We found for example an APC W2658L mutation to reduce the binding to both CASK and to GABARAPL1 based on the GenVar_HD2 selection results. Alternatively, a mutation may affect the binding of one domain to one SLiM, while having no impact on other interactions. For example, we found that the SQSTM1 P348L mutation (334-GDDD<u>WTHL</u>SSKEVD(P/L)<u>STGEL</u>QSL-356) conferred a 25-fold decrease in affinity for KEAP1 KELCH domain but did not affect the binding to the MAP1LC3B ATG domain (Table 1). Alternatively, as demonstrated through affinity measurements for the CTNNB1 S33F, a mutation may simultaneously abrogate binding to one protein, while creating a novel binding site for another protein. In the cases where the perturbed motifs are located in distinct regions of the mutated protein, the mutation of one SLiM will not directly affect binding to the other proteins by the other SLiM. However, even if the binding sites and interactions per se are independent, there may be changes in the interactomes of the mutant protein. For example, the loss of an NLS will lead to cytosolic localisation of the protein, and hence loss of binding with nuclear interaction partners (e.g. CDC45). Thus, a single SNV of a SLiM may perturb a PPI network in many different ways.

The mutations of the GenVar_HD2 library originate from a diverse set of diseases, such as neurological diseases, metabolic diseases and cancer. We hence provide a disease-panoramic view of how disease-associated mutations enable/disable SLiM-based interactions. We found disruption and enhancement of interactions with E3 ligases as common features of various disease categories. Examples include the SCNN1B mutations disrupting the binding to NEDD4 and mutations in both NFE2L2 and SQSTM1 diminishing the interaction with KEAP1. Our findings thus support the notion that perturbation of E3 ligase targeting and associated altered protein levels is a possible molecular mechanisms underlying disease progression (Mészáros et al, 2017). Similarly, we found numerous perturbed interactions with proteins involved in transport and trafficking, as well as scaffolding, suggesting that protein mislocalisation may be a common consequence of mutations occurring in the IDRs. This is highlighted by the newly discovered interaction between CDC45 and KPNA7, for which we demonstrated that the R157C CDC45 mutation, associated with Meier-Gorlin syndrome, disrupts KPNA7 binding and results in the delocalisation of CDC45 from the nucleus. As the picture we paint is biased by the proteins used as baits, we foresee that expanding and broadening the screen will allow the identification of other cellular processes perturbed by disease-associated mutations in the IDRs.

Validation of the effect of the binding-enhancing mutations proved more challenging in the context of full-length proteins by co-immunoprecipitation than on the peptide level by biophysical affinity measurements. This may suggest that these interactions might only occur under specific conditions or cell lines not recapitulated in HeLa/HEK293. It can also be hypothesised that breaking an interaction is more easily achieved in the cellular context than creating a novel interaction, taking the competitive environment of endogenous, potentially binding-optimised ligands into account. This is supported by the current literature primarily reporting on disruptive mutations (Sahni et al, 2015; Fragoza et al, 2019). It can be envisioned that certain interactions require additional disease-specific dysregulation besides the disease-associated mutation. This could include, for example, more than one cancer-promoting mutation in the same or additional proteins in order to create the cellular environment in which they become relevant.

We explored the target development level of proteins engaging in mutation-modulated interactions and found that there are few drugs targeting the proteins that we found to engage in interactions modulated by disease-associated mutations. Our results could thus inform on novel therapeutically interesting strategies for personalised medicine such as inhibiting neo-interactions or modulating E3 ligase-substrate targeting interactions, which may be created or weakened by the mutation. While it is challenging to target SLiM-binding pockets, there is a continuous development in the field of peptide-based therapeutics (Wang et al, 2022: Simonetti et al, 2023). This includes, for example, proteolysis-targeting chimeras (PROTACs) (Bekes et al, 2022) but also the stabilisation of SLiM-based interactions by small molecules, as shown for the S37A mutant CTNNB1 and β-TrCP E3 ligase interaction (Simonetta et al, 2019). Small molecule inhibitors of SLiM-binding pockets may further be developed starting from the information on the binding motifs, as recently shown for the NTF2-like domain of G3BP1 (Freibaum et al, 2024).

In summary, mutational ProP-PD is an effective approach for characterising the consequence of disease-associated mutations on motif-mediated interactions on large-scale. This is achieved by screening thousands of mutation sites simultaneously in a single experiment. The main limitations of the study are the limited number of bait proteins screened and the number of interactions tested in the cellular context. Expanding the protein collection and pairing the approach with other large-scale interactomics approaches can thus be envisioned. We acknowledge that mutations occurring in the IDRs can have functional consequences apart from protein binding, for example by modulating the dynamics of IDRs, changing the phase separation propensity of low complexity regions or changing the PTM modification state of the full-length protein. With those limitations in mind, we find that mutational ProP-PD presents a powerful approach to establish novel functional relationships between mutations and protein binding on a large-scale, while concomitantly providing information on interaction partners and binding sites.

# Methods

### Reagents and tools table

| Reagent/Resource | Reference or source | Identifier or catalogue number |
|---|---|---|
| **Experimental Models** | | |
| Escherichia coli BL21(DE3) gold | Agilent Technology | Cat. 230132 |
| Escherichia coli OmniMAX | ThermoFisher | Cat. C854003 |
| Escherichia coli SS320 | Lucigen | Cat. 60512-1 |
| M13KO5 helper phage | ThermoFisher | Cat. 18311019 |
| HEK293T cells | Sigma | Cat. 85120602 |
| HeLa cells | ATCC | CCL2 |
| **Recombinant DNA** | | |
| pETM33 | EMBL | |
| pETM41 | EMBL | |
| Phagemid p8 | Sidhu lab (Chen (2015)) | |
| EGFP-C1 plasmid | Addgene (discontinued; Zimmermann (2001)) | |
| pcDNA/FRT/TO | ThermoFisher | Cat. V652020 |
| YFP pcDNA/FRT/TO | Nilsson lab - cloning YFP with HindIII in pcDNA5/FRT/TO | |
| **Antibodies** | | |
| HRP-conjugated anti-M13 bacteriophage monoclonal mouse antibody | Sino Biological Inc | Cat. 11973-MM05T-H / RRID: AB_2857928 |
| anti-c-Myc Monoclonal Antibody (9E10) mouse | Thermo (Fisher Scientific) | Cat. 13-2500 / RRID:AB_2533008 |
| T7-Tag (D9E1X) XP® Rabbit mAb | Cell signalling | Cat. 13246 / RRID:AB_2798161 |
| Monoclonal ANTI-FLAG® M2 antibody produced in mouse | Sigma | Cat. F3165 / RRID:AB_259529 |
| anti-GFP (mouse/rabbit) antibody | rabbit- anti-GFP, produced in-house or mouse anti-GFP from Roche | Cat. 11814460001 / RRID: AB_390913 for Roche |
| IRDye® 800CW Goat anti-Mouse IgG Secondary Antibody | LI-COR | Cat. 926-32210 / RRID: AB_621842 |
| IRDye® 680RD Goat anti-Rabbit IgG Secondary Antibody | LI-COR | Cat. 926-68071 / RRID:vAB_10956166 |
| **Oligonucleotides and sequence-based reagents** | | |
| Primers | Details in Dataset EV11. Eurofins. | |
| Oligonucleotides GenVar_HD2 | CustomArrray | |
| **Chemicals, enzymes and other reagents** | | |
| 0.25% Trypsin/EDTA | Thermo (Fisher Scientific) | Cat. 25200-056 |
| DMEM GlutaMax 4.5 g/L D-Glucose, Pyruvate | Thermo (Fisher Scientific) | Cat. 31966047 |
| OPTIMEM medium | Invitrogen | Cat. 51985026 |

| Reagent/Resource | Reference or source | Identifier or catalogue number |
|---|---|---|
| Anti-FLAG® M2 Magnetic Beads | Merck (Millipore) | Cat. M8823 |
| GFP-Trap Agarose | Chromotek | Cat. GTA-20 |
| PhosSTOP Easypack | Roche | Cat. 04906837001 |
| IGEPAL CA-630 | Sigma | Cat. I8896 |
| Protein Assay Dye Reagent Concentrate | Bio-Rad | Cat. 5000006 |
| NuPAGE LDS Sample buffer (4x) | Novex | Cat. NP0008 |
| MOPS SDS Running Buffer (20x) | Invitrogen | Cat. NP0001-02 |
| cOmplete™ EDTA-free Protease Inhibitor Cocktail | Roche | Cat. 4693132001 |
| ProLong™ Glass Antifade Mountant with NucBlue™ Stain | Invitrogen | Cat. P36983 |
| T4 polynucleotide kinase | Thermo (Fisher Scientific) | Cat. EK0031 |
| T7 DNA polymerase | Thermo (Fisher Scientific) | Cat. EP0081 |
| T4 DNA ligase | Thermo (Fisher Scientific) | Cat. EL0014 |
| 50 bp marker | Thermo (Fisher Scientific) | Cat. 10416014 |
| Mag-bind Total Pure NGS | Omega Bio-tek | Cat. M1378-01 |
| QIAquick Gel extraction Kit | Qiagen | Cat. 28706×4 |
| Quant-iT PicoGreen dsDNA Assay Kit | Molecular probes by Life Technologies | Cat. P7589 |
| TMB substrate | Seracare KPL | Cat. 5120-0047 |
| QIAquick Nucleotide Removal Kit | Qiagen | Cat. 28306 |
| GSH Sepharose 4 Fast Flow Media | Cytiva | Cat. 17513201 |
| Ni Sepharose excel | Cytiva | Cat. 17371201 |
| Affinity chromatography columns, HiTrap™ Benzamidine FF | Cytiva | Cat. 17514302 |
| Thrombin | Cytiva | Cat. 27-0846-01 |
| Thermo Scientific™ Lysozyme | ThermoFisher | Cat. 89833 |
| GelRed | Biotium | Cat. 41003-T |
| Thermo Scientific™ Phusion High-Fidelity PCR Master Mix with HF Buffer | ThermoFisher | Cat. F631L |
| QIAquick PCR Purification Kit | Qiagen | Cat. 28104 |
| MinElute PCR Purification Kit | Qiagen | Cat. 28004 |
| jetOPTIMUS® DNA transfection reagant and buffer | Polyplus | Cat. 101000006 |
| 96-well Flat-bottom Immunosorp MaxiSorp plates | Nunc, Roskilde, Denmark | Cat. 439454 |
| 384-well Flat-bottom Immunosorp MaxiSorp plates | Nunc, Roskilde, Denmark | Cat. 464718 |

| Reagent/Resource | Reference or source | Identifier or catalogue number |
|---|---|---|
| 96-well half-area black Flat-bottom Non-binding surface plates | Corning, USA | Cat. 3993 |
| Nunc EasYFlask 175cm2 | Nunc, Roskilde, Denmark | Cat. 159920 |
| 145×20 cm Greiner Cellstar Cell Culture Dish | Greiner Bio-One | Cat. 639160 |
| Nunc™ Lab-Tek™ II CC2™ Chamber Slide System 8-well | Thermo (Fisher Scientific) | Cat. 154941PK |
| **Software** | | |
| GraphPad Prism version 9.5.1 for MacOS | GraphPad Software, San Diego, California USA, www.graphpad.com | |
| ImageJ | https://imagej.nih.gov/ij/ | |
| PyMOL Version 2.1.1 | New York, New York, USA Schrodinger LLC | |
| ZEN Microscopy Software | Zeiss | |
| Python version 3 | New York, NY, USA Schrodinger LLC | |
| Pandas | McKinney (2010) | |
| Matplotlib | Hunter (2007) | |
| Seaborn | Waskom et al (2021) | |
| R | https://www.r-project.org | |
| R studio | http://www.rstudio.com/. | |
| Affinity designer for MacOS | https://affinity.serif.com/en-us/designer/ | |
| Inkscape | Inkscape Project. (2020). Inkscape. Retrieved from https://inkscape.org | |
| **Other** | | |
| iD5 | Molecular Devices | |
| iTC200 | Malvern | |
| Illumina MiSeq v3 run, 1 × 150 bp read setup, 20% PhiX | NGS-NGI SciLifeLab facility | |
| LSM700 confocal microscope | Zeiss | |

## Peptides

Peptides were synthesised by and bought from Genecust, France (Dataset EV8). FITC-labelled peptides were dissolved in 200 μL DMSO, whereas unlabelled peptides were dissolved in 500–550 μL of 50 mM sodium phosphate buffer pH 7.4. On occasion, unlabelled peptide stocks were supplemented with 1–5 μL of 1 M NaOH to improve solubility. Further, resuspended unlabelled peptides were spun down for aggregates at $10,000 \times g$ for 10 min at 4 °C. If required, peptides were modified to contain an N- or C-terminal tyrosine to allow concentration determination by measuring absorbance at 280 nm.

## Protein domain expression and purification

Either Rosetta (kinase domains) or BL21 DE (Gold) (remainder of domains) *Escherichia coli* were used as host strains for protein

expression. Transformed bacteria were grown in 2YT (5 g/L NaCl, 16 g/L tryptone and 10 g/L yeast extract) at 37 °C at 220 rpm until they reached OD = 0.7–0.8. Protein expression was induced with 1 mM isopropyl β-ᴅ-1-thiogalactopyranoside (IPTG) and the bacteria incubated afterwards for 16–18 h at 18 °C and 220 rpm. Bacteria were spun down (15 min, $4000–5000 \times g$) and the pellets were stored at −20 °C until purification. For the phage display experiments, protein domains were used as tagged constructs (either glutathione transferase (GST)- or maltose binding protein (MBP)-tagged), whereas most domains were cleaved for affinity measurements. Bacteria pellets were resuspended in lysis buffer (0.05% Triton-X, 5 mM $MgCl_2$ in 1x phosphate-buffered saline (PBS) supplemented with 10 μg/mL DNase, cOMplete EDTA-free protease inhibitor (Roche) and lysozyme) and incubated for 1 h in the cold room while shaking and in the meantime sonicated. After that, the sample was centrifuged for 1 h at $16,000 \times g$ and 4 °C and the supernatant was incubated with either $Ni^{2+}$ (MBP- and 6xHis-tagged constructs; Cytiva) or glutathione (GST-tagged constructs; Cytiva) beads for 1–2 h in the cold room while shaking. Next, the beads were washed with either 1xPBS (glutathione beads) or $Ni^{2+}$washing buffer (40 mM imidazole, 250 mM NaCl, 20 mM sodium phosphate pH 7.4 washing buffer; $Ni^{2+}$ beads) respectively. Protein domains were eluted with either 10 mM glutathione (glutathione beads) or $Ni^{2+}$ elution buffer (300 mM imidazole, 250 mM NaCl, 20 mM sodium phosphate pH 7.3; $Ni^{2+}$ beads) and afterwards supplemented with 10% glycerol and stored at −80 °C. For affinity measurements, all constructs, except for NEDD4 WW2-GST and 6xHis constructs, were cleaved from their tag proteins. Cleavage was performed overnight either with the human Rhinovirus (HRV) 3 C protease in 50 mM Tris HCl pH 8.0, 150 mM NaCl, 1 mM DTT or thrombin in 1xPBS with 1 mM DTT. For HRV 3C protease cleavage, the proteins were either eluted from the glutathione beads and dialysed into the cleavage buffer, or purification and cleavage was on $Ni^{2+}$ beads, so that the pure protein was in the flow-through after cleavage. In the first approach, cleaved GST and the protease were removed by reverse $Ni^{2+}$ purification (50 mM Tris HCl, pH 8.0). For thrombin cleavage, cleavage was on the glutathione beads and the protease was removed with a benzamidine column (Cytiva). The protein domains for affinity measurements were dialysed in 50 mM sodium phosphate buffer with 1 mM DTT. For the CASK kinase domain, the buffer was additionally supplemented with 100 mM NaCl.

## GenVar_HD2 library design

The GenVar_HD2 library design was built on top of our previously published HD2 library (Benz et al, 2022) by introducing pathogenic or likely pathogenic mutations from publicly available databases (gnomAD, dbSNP, COSMIC curated, TOPMed, NCI-TCGA COSMIC, NCI-TCGA, ExAC, Ensembl, ESP, ClinVar, 1000 Genomes retrieved from the EBI API (Nightingale et al, 2017), and UniProt Human polymorphisms and disease mutations (UniProt: a worldwide hub of protein knowledge, 2019) as of December of 2019). To avoid disulphide bonds all cysteines were replaced by alanines both in the wild-type peptide and the mutant sequences. Almost all mutations were single nucleotide variants (SNV) with two exceptions (P50461.S54_E55delinsRG and P51587.T298_V300delinsILR). Only a single mutation per designed peptide was allowed (either a SNVs or one of the two multi-site

aforementioned mutations), so in the case where multiple mutations could occur in the same peptide region, one peptide per mutation was generated. All wild-type and mutant peptides were then reverse-translated into oligonucleotides with optimised codons for expression in *E. coli* avoiding the generation of SmaI restriction sites. Most mutations in the resulting design are covered by 3 or more overlapping peptides.

## Generation of the GenVar_HD2 library

The GenVar_HD2 library was generated on the major coat protein p8 of the M13 bacteriophage according to previous protocols (Benz et al, 2022; Ali et al, 2020; Kliche et al, 2023), which included the amplification of the oligonucleotide pool, subsequent annealing to the single-stranded phagemid and its amplification in a Kunkel reaction. The library was then transformed into *E. coli* SS320 bacteria by electroporation. The procedure was in brief as follows. The oligonucleotide library (CustomArray) was used as a template in a PCR reaction run for 18 cycles with the Phusion High-Fidelity PCR Master Mix (Thermo Scientific) and amplification was confirmed by running the final product on 2% agarose gel. The PCR reaction was cleaned up by the MinElute PCR Purification Kit (Qiagen) and subsequently phosphorylated for 1 h at 37 °C with the T4 polynucleotide kinase (10 units; Thermo Scientific). Annealing of the phosphorylated PCR product to the single-stranded p8 phagemid was performed by a temperature gradient of 90 °C (3 min), 50 °C (3 min) and 20 °C (5 min). The resulting dsDNA was next amplified with the help of a T4 DNA ligase (30 Weiss units; Thermo Scientific) and a T7 DNA polymerase (30 units; Thermo Scientific) for 4 h at 20 °C. The product was cleaned up on QIAquick spin columns (Qiagen) and assessed by running it on a 1% agarose gel. The generated dsDNA was electroporated into *E. coli* SS320 bacteria (Lucigen), pre-infected with M13K07 helper phage (Thermo Scientific) (Rajan and Sidhu, 2012). The cells were recovered in pre-warmed SOC medium (20 g/L tryptone, 5 g/L yeast extract, 10 mM NaCl and 2.5 mM KCL pH 7.0; added after autoclaving: 10 mM MgCl$_2$ and 20 mM glucose) and grown for 24 h in 2YT medium. Bacteria were pelleted by centrifugation (4000 × *g*) and the phages precipitated by the addition of one-fifth of PEG-NaCl (20% w/v PEG-8000 and 2.5 M NaCl) and 5–10 min incubation on ice. This was followed by centrifugation at 16,000 × *g* for 20 min at 4 °C to collect the phages, which were then resuspended in 1xPBS with 0.05% Tween-20 and 0.2% BSA (bovine serum albumin). Insoluble debris was removed by spinning at 27,000 × *g* for 20 min, and the final phage library was supplemented with 10% glycerol and stored at −80 °C.

## Phage display

Phage display screens were performed according to previous reports (Benz et al, 2022; Ali et al, 2020) and for four consecutive selection rounds with five to six replicates per bait protein domain against the GenDi_HD2 library.

### Selection day 0

On the first day, both bait (15 µg in 100 µL PBS) and control protein domains (GST and MBP; 10 µg in 100 µL PBS) were immobilised on a 96-well MaxiSorp plate (Nunc) overnight in the cold room while shaking. Additionally, a 10 mL culture of

Omnimax *E. coli* bacteria supplemented with 10 µg/mL tetracycline was incubated overnight at 37 °C at 200 rpm.

### Selection day 1

First, protein solutions were removed from both control and bait protein domain plates and the wells were blocked with 0.5% BSA in PBS (200 µL/well) for 1 h in the cold room while shaking. After that, three Omnimax *E. coli* cultures were started supplemented with either 30 µg/mL kanamycin, 100 µg/mL carbenicillin and 10 µg/mL tetracycline and incubated at 37 °C and 200 rpm. The two first cultures served as a means to check for helper phage contamination (kanamycin) and phages with library phagemid contamination (carbenicillin) of the culture used for elution of the phages during selections. The tetracycline Omnimax *E. coli* culture was grown until OD at 600 nm reached 0.6–0.8.

Naïve phage libraries were precipitated (10 µL/well of bait protein domain) by diluting the library tenfold with PBS, adding one-fourth of 20% PEG-800 and 400 mM NaCl and incubating the mixture for 15 min on ice. Subsequently, phages were spun down for 15 min at 13,000 × *g* and resuspended in 0.5% BSA and 0.05% Tween-20 in PBS (100 µL/well). Control domain plates were washed four times with 200 µL 0.05% Tween-20 in PBS, 100 µL of the naïve phage library (10$^{11}$ phages/mL) were added to every well and the plates incubated for 1 h in the cold room while shaking. Next, protein bait domain plates were washed four times with 0.05% Tween-20 in PBS and the phage solution from the control protein plate was transferred to the respective wells on the protein bait domain plate. The protein bait domain plate was incubated with the phages for 2 h in the cold room while shaking, allowing for the binding of the peptides on the phages to the bait protein domains. After that, the bait protein domain plate was washed five times with 0.05% Tween-20 in PBS and the bound phages were eluted with the log-phase Omnimax *E. coli* (100 µL/well) by incubating the plates for 30 min at 37 °C at 200 rpm. Then, the bacteria were infected with M13K07 helper phages (100 µL of 10$^{12}$ phages/mL) and incubated for 45 min at 37 °C at 200 rpm. Phages were amplified by growing the infected bacteria in 96-well deep well plates with 2YT supplemented with 30 µg/mL kanamycin, 100 µg/mL carbenicillin and 0.3 mM IPTG overnight at 37 °C and 200 rpm. Last, control and bait protein domains were immobilised in the same fashion as on Day 1 on 96-well MaxiSorb plates (Nunc).

### Selection days 2–4

The procedure for selection days 2–4 were the same as for selection day 1 with the following exception. The amplified phages from the previous selection round were harvested by first spinning down the bacteria for 15 min at 1700 × *g* at 4 °C and transferring the phage supernatant (800 µL) to a new 96-deep-well plate, in which 80 µL of 10x PBS have prior been added to every well. Those plates were incubated for 10 min at 60 °C and afterwards cooled down on ice. The phage supernatants were subsequently used as in-phages for the next selection round and added to the respective well on the washed control domain plate (100 µL/well). In addition, the supernatants were used in the phage pool ELISA (enzyme-linked immunosorbent assay) or for next-generation sequencing.

## Phage pool ELISA

The protocol of the phage pool ELISA experiment was followed as previously reported and was performed for the four selection days,

occasionally only for selection days 3 and 4. Shortly, control (5 µg) and bait (7.5 µg) protein domains were added in 50 µL PBS to a 384-well MaxiSorb plate (Nunc) and immobilised overnight in the cold room while shaking. On the next day, the wells were blocked with 100 µL 0.5% BSA in PBS and incubated for 1 h in the cold room while shaking. The blocking solution was removed and phage supernatant from the respective selection days and bait protein domain were added to both the immobilised control and bait protein domain well. The plate was subsequently incubated for 1–2 h in the cold room while shaking. As the next step, the plate was washed four times with 0.05% Tween-20 in PBS and 100 µl of anti-M13 HRP-conjugated antibody (Sino Biological Inc; 1:5000 diluted in 0.5% BSA, 0.05% Tween-20 in PBS) were added to each well, which was followed by incubation for 1 h in the cold room while shaking. The plates were then washed four times with 0.05% Tween-20 in PBS and one time with PBS only, before adding 40 µL of TMB substrate (Seracare KPL) to each well. The enzymatic reaction was then stopped by the addition of 40 µL of 0.6 M $H_2SO_4$. The absorbance at 405 nm was subsequently measured on an iD5 plate reader (Molecular Devices) as read-out of the assay.

## NGS

Barcoding and NGS was performed according to previously established protocols (Benz et al, 2022; Ali et al, 2020; Kliche et al, 2023; McLaughlin and Sidhu, 2013). The peptide-coding regions of the naïve library and the enriched phage pools were barcoded using a dual barcoding strategy. This was achieved by PCR amplification (5 µL template; 22 cycles) using custom-made primers and the Phusion High-Fidelity PCR Master Mix (Thermo Scientific). Amplification was confirmed on a 2% agarose gel and the PCR products (25 µL) were purified with 25 µL Mag-bind Total Pure NGS beads (Omega Bio-tek). Concomitantly, samples were normalised by taking 10 µL of each sample during the elution from the beads (Qiagen Elution buffer: 10 mM Tris-Cl and 0.1 mM EDTA (pH 8.5)) and subsequently pooled. The pooled samples were cleaned-up with a PCR purification kit (Qiagen) and subsequently run on a 2% agarose gel to extract bands of the correct size (ca. 200 bp) with the help of a QIAquick Gel extraction Kit (Qiagen). The samples were eluted with 30 µL TE buffer (10 mM Tris HCl, 1 mM EDTA. pH 7.5) buffer and quantification was performed with Quant-iT PicoGreen dsDNA Assay Kit (Molecular probes). Samples were pooled and sent for sequencing to the NGS-NGI SciLifeLab facility, Solna, Sweden. Sequencing was on MiSeq (MSC 2.5.0.5/RTA 1.18.54) with a 151 nt (Read1) setup using 'Version3' chemistry. The Bcl to FastQ conversion was performed using bcl2fastq_v2.20.0.422 from the CASAVA software suite.

## Processing of NGS data

The processing of the NGS sequencing data was performed as described previously (Benz et al, 2022). Pooled experimental sequences were demultiplexed by identifying unique sets of 5′ and 3′ barcodes. Sequences with an average quality score of less than 20 were discarded. Adaptor regions were trimmed, and finally, the DNA sequences were translated to amino acids using in-house custom Python scripts. Results for each experimental selection were compiled containing peptide sequences together with their observed NGS counts, and these tables were annotated and analysed utilising the PepTools pipeline (http://slim.icr.ac.uk/tools/peptools/).

## Library coverage and quality

To evaluate the quality and completeness of the GenVar library, multiple unchallenged naïve aliquots of the phage library (5 µL) were barcoded and sequenced as described in the previous sections. The observed coverage was calculated at the peptide level together with a predicted maximum coverage as described in Benz et al, (Benz et al, 2022) (Appendix Fig. S1A), and the completeness of the library wild-type/mutant pairs in the library was assessed (Appendix Fig. S1B,C).

### Selection results analysis

Individual NGS, selection results tables were processed as described by Benz et al, (Benz et al, 2022). Results were cleaned by removing peptides with a count of 1 and by removing any peptide that did not match the library design. Different selection days for the same experiment were merged and a *per* peptide normalised average count was calculated. Replicated results for the same bait were analysed together: First, peptide sequences were annotated and motif identification was carried out with the PepTools pipeline (http://slim.icr.ac.uk/tools/peptools/), and then a confidence score was assigned to each identified peptide based on (i) occurrence in the replicate selection, (ii) peptide overlap, (iii) motif matching and (iv) NGS counts, resulting in peptides having a final score between 0 and 4 depending on how many of the criteria they met. For simplification, peptides with a score of 2 or 3 are considered medium-confidence while those with a score of 4 are high confidence.

### Enrichment analysis of previously reported interactions

To establish if there was a significant enrichment of previously reported interactions, we first updated our PPI datasets for human proteome (December-2023). Our PPI datasets include experimentally validated protein–protein interactions obtained from the IntAct (DelToro et al, 2022), HIPPIE (Alanis-Lobato et al, 2017), HURI (Luck et al, 2020), Bioplex (Huttlin et al, 2021) and STRING (Szklarczyk et al, 2023) databases. Next, we calculated the total number of possible unique protein–protein pairs, excluding self-binding, considering all baits used in the study and all preys in the GenVar library design (total: 147,147 PPIs). Next, we annotated how many of those protein–protein pairs have been characterised in our PPI dataset (known PPIs: 4638). We observed that from 147,147 possible protein–protein pairs, 4638 have been reported as interacting proteins in databases, and we used it as a background in our comparison to our selection results. In the current study, we observed 1229 possible unique high-/medium-confidence interactions at the protein–protein level, from which 100 were previously reported as experimentally validated PPIs in the aforementioned databases. Based on observed values in this study (100 known out of 1229) compared to the background library settings (4638 known out of 147,147), we obtained an enrichment of 2.58.

### Mutation-centred analysis

A mutation-centred analysis was performed by comparing wild-type and mutant peptide counts as described in Kliche et al, (Kliche

et al, 2023). In brief: Two metrics were calculated for the wild-type and mutant overlapping peptides for each mutation found for a bait: a Mann–Whitney confidence score and a mutation enrichment score.

$$\text{Mutation enrichment score} = \sum_{i=0}^{n} \frac{nc^i_{mut}}{nc^i_{wt} + nc^i_{mut}} - \sum_{i=0}^{n} \frac{nc^i_{wt}}{nc^i_{wt} + nc^i_{mut}}$$

Where $nc$ are the normalised counts for the wild-type (wt) or mutant (mut) peptides, and $n$ corresponds to the number of overlapping peptide pairs found for a wt/mut position in all selection replicas. A positive score indicates a mutation enhancing binding while a negative value indicates a mutation disrupting binding.

## Motif consensus mapping

In order to map the motif instances of the bait protein domain, amino acid stretches flanking the mutation site (±20 residues) of the prey protein were used. First, previously known motif instances were obtained by mapping experimentally validated motifs derived from either curated literature, previously published ProP-PD results (Benz et al, 2022; Kruse et al, 2021; Kliche et al, 2023) or the ELM database (Kumar et al, 2022). For the mutation sites without any known overlapping motif instance, we used motif consensus and PSSM models to search the region to predict the putative motif instance in the mutation region. SLiMSearch (Krystkowiak and Davey, 2017) and PSSMSearch (Krystkowiak et al, 2018) were applied for motif consensus and PSSM model prediction, respectively (disorder cut-off = 0, remaining options default). The motif mapping was limited to the previously known and predicted motif instances of the bait protein domain found in the GenVar_HD2 selections to bind to wild-type and/or mutant peptides from the prey protein. This entails that motif mapping was performed as a pair of the bait protein domain and the mutation site of the prey protein. The expected motif instances for the bait protein domains are listed in Dataset EV2.

Next, the mutation sites were mapped on the motif region to determine their position in relation to the motif instance. In this fashion, we define four mutation categories when a motif instance is found: mutations occurring in key residues, wild-card and flanking positions, as well as when the mutation creates the motif instance (Dataset EV6). The key residue and wild-card positions of a motif instance were derived from the motif consensus representation, where wild-card positions allow any amino acid and the key positions define specific amino acids or groups of amino acids in the motif consensus. In addition, we report on the closest motif match to the mutation site if there was no motif directly overlapping the mutation site (i.e. the mutation is found in the flanking region of the motif instance). Furthermore, if the motif instance can only be found in the mutant peptide (not in the wild-type peptide) and if the mutated residue overlaps with the key residues of the motif instance, we consider them as neo-motif as the mutation created the motif instance. Of those four categories, we consider mutations in key residues to impair the motif function, since there is no longer a match for the motif consensus, and mutations creating the instance to gain the motif function.

## Fluorescence polarisation (FP) experiments

FP experiments were performed, as previously described (Kliche et al, 2021, 2023), on an iD5 plate reader (Molecular devices) using black half-area non-binding 96-well plates (Corning). In order to calculate the FP signal in millipolarisation (mP), the FITC-labelled peptides were excited at 485 nm and the emitted light was measured at 535 nm in two different angles. For saturation experiments, a dilution series of the respective protein domains was generated (25 µL) and 10 nM fluorescein isothiocyanate (FITC)-labelled peptide (diluted in 50 mM sodium phosphate buffer pH 7.4, 0.05% Tween) were added in an end volume of 50 µL. A control experiment titrating GST was performed for the binding of the NEDD4 WW2 domain, since the GST-tagged construct was used for affinity measurements. For fitting the data, the buffer-corrected reduced values were plotted in GraphPad PRISM against the respective protein concentration and the $K_D$-values were obtained by fitting against the quadratic equation (Gianni et al, 2005).

In turn, for displacement experiments, a dilution series of the unlabelled peptides (25 µL) was generated and 25 µL of a complex of FITC-labelled peptide (10 nM) and protein domain (concentration = 1-2x $K_D$) (25 µL) was added. If the raw values indicated a FITC-contamination of the unlabelled peptide, the data was excluded and a new batch of peptide was ordered for remeasuring the affinity. The buffer-corrected reduced values were subsequently plotted against the logarithmic unlabelled peptide concentration to derive the IC50 values by using the sigmoidal dose-response (x is logarithmic) fit in GraphPad PRISM and calculate the $K_D$ values as described previously (Nikolovska-Coleska et al, 2004).

## Isothermal titration calorimetry experiments

For the isothermal titration calorimetry experiment, both MAP1LC3B ATG8 and BRCA2$_{292-307}$ peptides were dialysed in 50 mM sodium phosphate buffer pH 7.4, 1 mM DTT to avoid buffer mismatch. An iTC200 instrument (Malvern) was used and the peptides were titrated in tenfold excess in 16 injections against the protein domain. Curve fitting and baseline correction were done with the programme on the instrument. The wild-type peptide was measured in a technical duplicate and the mutant peptide in technical triplets.

## Co-immunoprecipitation experiments

For Co-immunoprecipitation experiments, selected proteins were YFP- (MAP1LC3B, CSKP and G3BP1), EGFP- (ABRAXAS1 and CDC45), GFP-(KEAP1) and the corresponding target proteins (MAPT, CTNNB1 and SQSTM1) were MYC-tagged. Additionally, the target proteins and EGFP-CDC45 were mutated by site-directed mutagenesis to introduce the mutations observed in the phage display. The FLAG-ABRAXAS1 wild-type and mutant constructs used for cloning in an EGFP-C1 vector were a kind donation from Prof. Robert Winqvist. The T7-KPN (1-4,6,7) constructs were purchased from addgene and Myc-CDC45 from origene.

HeLa and HEK293T cells were maintained in DMEM, high glucose, GlutaMAX™ Supplement (Gibco) supplemented with 10% FBS and 1% penicillin/streptomycin at 37 °C and 5% $CO_2$. Cells

were co-transfected with the wild-type (1–2 μg) or mutant (1–2 μg) prey protein plasmid and either YFP only (0.5–1 μg) or the YFP-tagged bait protein plasmid (1–2 μg). Exceptions were transfections with EGFP-ABRAXAS1/-CDC45, where 4 μg of plasmid were used for the EGFP-construct and T7-KPN and 1 μg of EGFP plasmid. Then, plasmids were mixed with 0.5–1 mL jetOPTIMUS buffer (Polyplus) and 2–8 μL of jetOPTIMUS transfection reagent (Polyplus, 1 μL/μg plasmid) were added. The samples were briefly vortexed, spun down and incubated at room temperature for 15 min. Subsequently, the mixtures were added directly to the medium of a 15 cm dish (Greiner) or 175-cm² flask (Nunc). The medium was changed 6–18 h after transfection and the cells were collected 24 or 48 h after transfection by trypsinisation.

The cells were lysed by resuspending the cell pellet in 0.5–0.8 mL lysis buffer (100 mM NaCl, 50 mM Tris pH 7.4, 0.05% NP-40 (Igepal), 1 mM DTT, cOMPLETE EDTA-free protease and phos-stop phosphatase inhibitors) and sonicating them for 10 min (ten cycles, 30 pulse, 30 s pause) at 4 °C (Bio-ruptor). Exceptions were the KPN Co-IPs, which were instead incubated for 45 min on ice. The samples were subsequently spun for 45–60 min $>15,000 \times g$ and the supernatant was added to 20 μL of GFP-Trap slurry (ChromoTek) in low-binding tubes. Binding to the GFP-Trap was allowed by incubating the samples for 1 h at 4 °C. Next, the GFP-Trap was washed by spinning down for 3 min at 3000 rpm and adding 1 mL of washing buffer (150 mM NaCl, 50 mM Tris pH 7.4, 0.05% NP-40 (Igepal), 5% glycerol, 1 mM DTT) after removing the supernatant. This was repeated three times. After that, the GFP-Trap was eluted by adding 20 μL of 2x loading dye (NuPAGE) and the samples were stored at −20 °C. For western blotting, the samples were first separated by SDS PAGE using PageRuler™ Prestained Protein Ladder (Thermo Scientific) and NuPAGE™ MOPS SDS Running Buffer (Thermo Scientific), and they transferred to Immobilon-FL membranes (Merck) by blotting for 3 h at 250 mA or 16 h at 80 mA (KPN Co-IP) in the cold room. Subsequently, the membranes were blocked with 5% milk in 1xPBS with 0.1% Tween-20. For visualisation, mouse anti-Myc antibody (1:1000, Thermo Scientific), mouse anti-Flag antibody (1:5000, Sigma Aldrich), rabbit anti-T7 (1:1000, Cell signalling), rabbit anti-GFP antibody (1:5000 in-house) and mouse anti-GFP (1:1000, Roche) antibody were used and the membranes incubated overnight at 4 °C with the antibodies in 2.5% milk in 1xPBS with 0.1% Tween-20. The membranes were washed three times for 5 min with 0.1% Tween in 1xPBS and then incubated with secondary antibody (Goat anti-mouse 680/800 or goat anti-rabbit 680/800, 1:5000; LI-COR) in 5% milk in 1xPBS with 0.1% Tween-20 for 1 h at room temperature. The membranes were then again washed three times for 5 min within 1xPBS with 0.1% Tween-20 and then visualised reading at 700 or 800 nm, respectively. If the membrane was retried for a second antibody, it was first blocked again with 5% milk in 1xPBS with 0.1% Tween-20 for 1 h at room temperature and then incubated with the primary antibody as described above.

## Nuclear localisation experiments

HEK293T cells were transfected with 0.25 μg of respective plasmid (EGFP-ABRAXAS1 and EGFP-CDC45 wild-type and R157C mutant) while seeding (ca. 25% confluency) in eight chamber slide system (Nunc Lab-Tek II CC2). The medium was changed after 24 h, and protein expression was allowed for 48 h in total after transfection. After that, cells were fixed and mounted with cell nuclei stained with NucBlue Live ReadyProbes (Invitrogen). Z-stacks of the cells were recorded on an LSM700 confocal microscope (Zeiss) and images analysed with Fiji.

## Data availability

The protein interactions from this publication have been submitted to the IMEx (http://www.imexconsortium.org) consortium through IntAct (Del Toro et al, 2022) and assigned the identifier IM-30020. ProP-PD results are also made available through the ProP-PD portal (https://slim.icr.ac.uk/proppd/proppd_data_viewer?table_type=bait_index&library=GenDi). The source data for Fig. 5H,I has been submitted to Biostudies (https://www.ebi.ac.uk/biostudies/) and has been assigned the BioImages accession number S-BIAD1223.

The source data of this paper are collected in the following database record: biostudies:S-SCDT-10_1038-S44320-024-00055-4.

## Peer review information

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

## Acknowledgements

This work was supported by grants from the Swedish Research Council (YI: 2020-03380), the Ollie and Elof Ericsson foundation (YI) and the Cancer Research UK (CRUK) (NED: Senior Cancer Research Fellowship C68484/ A28159). MD is supported by a Marie Sklodowska-Curie European Training Network Grant #860517 (Ubimotif). Sequencing was performed by the SNP&SEQ Technology Platform in Stockholm. The facility is part of the National Genomic Infrastructure (NGI) Sweden and Science for Life Laboratory and is also supported by the Swedish Research Council and the Knut and Alice Wallenberg. Work at the Novo Nordisk Foundation Center for Protein Research is supported by grants NNF14CC0001 and NNF23OC0082227. We thank Dr. Sachdev Sidhu for providing the phagemid and several of the expression constructs. We thank Prof. Robert Winqvist for providing the FLAG-ABRAXAS1 wild-type and mutant constructs, Dr. Andreas Ernst for the ATG8 domain constructs, and Prof. Carlos Fontes and Renaud Vincentelli for constructs for PDZ domain expression.

## Author contributions

**Johanna Kliche**: Conceptualisation; Data curation; Investigation; Visualisation; Writing—original draft; Writing—review and editing. **Leandro Simonetti**: Data curation; Investigation; Visualisation; Writing—review and editing. **Izabella Krystkowiak**: Software; Methodology; Writing—review and editing. **Hanna Kuss**: Investigation. **Marcel Diallo**: Investigation. **Emma Rask**: Investigation. **Jakob Nilsson**: Supervision; Funding acquisition; Writing—review and editing. **Norman E Davey**: Conceptualisation; Software; Funding acquisition; Writing—review and editing. **Ylva Ivarsson**: Conceptualisation; Supervision; Funding acquisition; Writing—original draft; Writing—review and editing.

Source data underlying figure panels in this paper may have individual authorship assigned. Where available, figure panel/source data authorship is listed in the following database record: biostudies:S-SCDT-10_1038-S44320-024-00055-4.

## Funding

## Disclosure and competing interests statement

The authors declare no competing interests.

