## [Peer Review File · Molecular Systems Biology]

Proteome-scale characterisation of motif-based interactome rewiring by disease mutations

Johanna Kliche, Leandro Simonetti, Izabella Krystkowiak, Hanna Kuss, Marcel Diallo, Emma Rask, Jakob Nilsson, Norman Davey, and Ylva Ivarsson

Corresponding author(s): Ylva Ivarsson (ylva.ivarsson@kemi.uu.se) , Norman Davey (norman.davey@icr.ac.uk)

Review Timeline:

Submission Date:	15th Sep 23
Editorial Decision:	10th Oct 23
Revision Received:	30th Apr 24
Editorial Decision:	27th May 24
Revision Received:	14th Jun 24
Accepted:	28th Jun 24

Editors: Maria Polychronidou and Poonam Bheda

Transaction Report:

10th Oct 2023

Manuscript Number: MSB-2023-11986

Title: Proteome-scale characterisation of protein motif interactome rewiring by disease mutations

Dear Ylva,

Thank you again for submitting your work to Molecular Systems Biology. We have now heard back from the three reviewers who agreed to evaluate your study. As you will see below, the reviewers are overall quite positive about the relevance of the study for the field. They do however raise a series of concerns which we would ask you to address in a revision.

I think that the issues raised by the reviewers are rather clear and seem straightforward to address. All issues raised need to be satisfactorily addressed. Please feel free to contact me in case you would like to discuss in further detail any of the concerns raised. I would be happy to schedule a call.

On a more editorial level, we would ask you to address the following points:

- The keywords need to be reduced to 5.
 - Please provide a .doc version of the manuscript text (including legends for the main figures) and individual production quality figure files for the main Figures (one file per figure). Tables should be included in the manuscript text (at the very end), together with their description/legend.
 - We have replaced Supplementary Information by the Expanded View (EV format). In this case, all additional figures can be included in a PDF called Appendix. Appendix figures should be labeled and called out as: "Appendix Figure S1, Appendix Figure S2... Appendix Table S1..." etc. Each legend should be below the corresponding Figure/Table in the Appendix. Please include a Table of Contents in the beginning of the Appendix. For detailed instructions regarding expanded view please refer to our Author Guidelines: .
 - Tables S1-S11 should be provided as Datasets EV1-EV11. Please provide one file per EV Dataset. Each file should include the description of the EV Dataset in a separate tab.
 - Please provide a "standfirst text" summarizing the study in one or two sentences (approximately 250 characters), three to four "bullet points" highlighting the main findings and a "synopsis image" (550px width and max 400px height, jpeg format) to highlight the paper on our homepage.
 - All Materials and Methods need to be described in the main text. We would encourage you to use 'Structured Methods', our new Materials and Methods format. According to this format, the Materials and Methods section should include a Reagents and Tools Table (listing key reagents, experimental models, software and relevant equipment and including their sources and relevant identifiers) followed by a Methods and Protocols section in which we encourage the authors to describe their methods using a step-by-step protocol format with bullet points, to facilitate the adoption of the methodologies across labs. More information on how to adhere to this format as well as downloadable templates (.doc or .xls) for the Reagents and Tools Table can be found in our author guidelines: . An example of a Method paper with Structured Methods can be found here:
 - Please include a "Disclosure and Competing Interests Statement" in the main text.
 - Please include a Data availability section describing how the data, code etc. have been made available. This section needs to be formatted according to the example below:
The datasets and computer code produced in this study are available in the following databases:
 - Chip-Seq data: Gene Expression Omnibus GSE46748 (<https://www.ncbi.nlm.nih.gov/geo/query/acc.cgi?acc=GSE46748>)
 - Modeling computer scripts: GitHub (<https://github.com/SysBioChalmers/GECKO/releases/tag/v1.0>)
 - [data type]: [full name of the resource] [accession number/identifier] ([doi or URL or identifiers.org/DATABASE:ACCESSION])
 - For data quantification: please specify the name of the statistical test used to generate error bars and P values, the number (n) of independent experiments (specify technical or biological replicates) underlying each data point and the test used to calculate p-values in each figure legend. The figure legends should contain a basic description of n, P and the test applied. Graphs must include a description of the bars and the error bars (s.d., s.e.m.).
 - When you resubmit your manuscript, please download our CHECKLIST (<https://bit.ly/EMBOPressAuthorChecklist>) and include the completed form in your submission.
- *Please note* that the Author Checklist will be published alongside the paper as part of the transparent process

(<https://www.embopress.org/page/journal/17444292/authorguide#transparentprocess>).

If you feel you can satisfactorily deal with these points and those listed by the referees, you may wish to submit a revised version of your manuscript. Please attach a covering letter giving details of the way in which you have handled each of the points raised by the referees. A revised manuscript will be once again subject to review and you probably understand that we can give you no guarantee at this stage that the eventual outcome will be favorable.

Kind regards,

Maria

Maria Polychronidou, PhD
Senior Editor
Molecular Systems Biology

We realize that it is difficult to revise to a specific deadline. In the interest of protecting the conceptual advance provided by the work, we recommend a revision within 3 months (8th Jan 2024). Please discuss the revision progress ahead of this time with the editor if you require more time to complete the revisions. Use the link below to submit your revision:

IMPORTANT: When you send your revision, we will require the following items:

1. the manuscript text in LaTeX, RTF or MS Word format
2. a letter with a detailed description of the changes made in response to the referees. Please specify clearly the exact places in the text (pages and paragraphs) where each change has been made in response to each specific comment given
3. three to four 'bullet points' highlighting the main findings of your study
4. a short 'blurb' text summarizing in two sentences the study (max. 250 characters)
5. a 'thumbnail image' (550px width and max 400px height, Illustrator, PowerPoint or jpeg format), which can be used as 'visual title' for the synopsis section of your paper.
6. Please include an author contributions statement after the Acknowledgements section (see <https://www.embopress.org/page/journal/17444292/authorguide>)
7. Please complete the CHECKLIST available at (<https://bit.ly/EMBOPressAuthorChecklist>).

Please note that the Author Checklist will be published alongside the paper as part of the transparent process (<https://www.embopress.org/page/journal/17444292/authorguide#transparentprocess>).

See also figure legend guidelines: <https://www.embopress.org/page/journal/17444292/authorguide#figureformat>

9. Please note that corresponding authors are required to supply an ORCID ID for their name upon submission of a revised manuscript (EMBO Press signed a joint statement to encourage ORCID adoption).

(<https://www.embopress.org/page/journal/17444292/authorguide#editorialprocess>)

Currently, our records indicate that the ORCID for your account is 0000-0002-7081-3846.

Link Not Available

The system will prompt you to fill in your funding and payment information. This will allow Wiley to send you a quote for the article processing charge (APC) in case of acceptance. This quote takes into account any reduction or fee waivers that you may be eligible for. Authors do not need to pay any fees before their manuscript is accepted and transferred to the publisher.

EMBO Press participates in many Publish and Read agreements that allow authors to publish Open Access with reduced/no publication charges. Check your eligibility: <https://authorservices.wiley.com/author-resources/Journal-Authors/open-access/affiliation-policies-payments/index.html>

*** PLEASE NOTE *** As part of the EMBO Press transparent editorial process initiative (see our Editorial at <https://dx.doi.org/10.1038/msb.2010.72>), Molecular Systems Biology publishes online a Review Process File with each accepted manuscripts. This file will be published in conjunction with your paper and will include the anonymous referee reports, your point-by-point response and all pertinent correspondence relating to the manuscript. If you do NOT want this File to be published, please inform the editorial office at msb@embo.org within 14 days upon receipt of the present letter.

Reviewer #1:

Summary

Kliche et al report on the results of a phage display screen for 80 motif-binding domains against against a peptide library that was designed to cover disordered regions in human proteins that carry missense mutations that are associated with disease. Of note, the peptide library contained wild type and mutant peptides enabling a mapping of mutations that would decrease or increase binding to any of the presented 80 domains. To the best of my understanding this is the largest and also most systematic investigation of the effect of disease mutations falling in short linear motifs (SLiMs) on the binding of folded domains. The authors report on close to 400 domain-mutation pairs that significantly altered binding based on a stringent cutoff of which about half increase and the other half decreased binding. Findings were further validated in orthogonal assays using fluorescence polarization with domains and peptides as well as colP with full length proteins and microscopy. The study suggests that alterations of domain-SLiM mediated protein interactions might contribute to the manifestation of a diverse set of genetic diseases. The study also provides first ideas for possible molecular mechanisms underlying the pathogenicity of reported mutations.

Minor points

page 5, third line: the grammar of this sentence does not seem right

page 5, section title: reveal, not reveals

page 5, second paragraph: I suggest rephrasing for clarity: We found that the peptide-containing prey proteins shared GO terms related to the GO categories subcellular localization, molecular function, and biological processes with ... (I had to read the sentence multiple times and look at the figure to understand what you were referring to.)

page 5, second paragraph towards the end: Your statement about the different number of interactors found for G3BP1/2 in the previous screen and the one reported here makes me wonder about the sensitivity of the screens and whether you can estimate the fraction of mutations screened whose effect you simply failed to detect due to limited sensitivity of the phage display screens. If you can estimate this, it would be nice to add it to the manuscript. If you cannot estimate it, it would still be nice to briefly discuss this in the discussion section.

page 6 section title: selection and not selections

page 7, second paragraph: There are more than 100 domain-mutation pairs for which no motif consensus sequence could be identified in the peptide. Quite some of these are also shown in the little networks drawn in Fig 4c-f and Fig 5c-d. How can they be interpreted? Do they point to extensions of the known motif consensus or alternative binding modes for the domains or to false positives? It would be nice if the authors were able to do more analyses on these or at least speculate on their interpretation in the discussion section.

page 8, first line of 2nd paragraph: the "that" is too much, write "which interactions are lost and ..."

page 8, 2nd paragraph: The authors report at the beginning of the section that interactions functioning in the ubiquitin system are more commonly lost by disease mutations while interactions functioning in the autophagy system are more often enabled, for example. The pie charts shown in Fig 4a I thought were not very helpful to visually support this statement and it certainly does not allow for a quantification. If possible, it would be nice to compute enrichment scores to support the statements.

page 8, 2nd paragraph: the reference to Fig 4b misses the 4

page 11, first couple of lines: Can you quantify your statement that no disease category was overrepresented in terms of mutations that showed an effect on domain-binding? Fig 6a is not very helpful to visually support this statement.

page 12, discussion: You state that "many of the deregulated interactions are thought to map to SLiM-based interfaces..." I would friendly disagree with this statement. I think that it is very unclear to which extent mutations in disordered regions are disease-causing because they for example fall into SLiMs. Many scientists are still uninformed about the many roles of disordered regions in proteins for protein function. As a consequence most mutations falling in disordered regions remain

uncharacterized while mutations falling in domains are more readily called pathogenic (this is based on own observations and knowing how variant effect predictors work). I would suggest to reconsider this statement and potentially rephrase. This is also exactly one of the highlights of the manuscript as a first experimental study that shows the potential of deregulation of SLiM-domain interactions in disease.

Fig 2e: The coloring of the wild-type vs mutant title of the figure confuses me. Aren't the blue and red dots representing enrichments or depletions of mutant vs wild-type peptide reads? If I am not mistaken then I would not color the title this way or change to "disabling vs enhancing mutations". I also find the log 10 on the y-axis non-intuitive. Wouldn't a log 2 make more sense for visualization purposes?

Fig 6b and 6c: Please add to the legend an explanation of the node shape. Why did you decide to show known interactions between the preys? I would have found it more interesting to see which of the bait-prey interactions were known before.

Fig 6d and 6e. I honestly don't find these last analyses very useful and helpful. If there is space limitations one could remove this last part.

I would encourage the authors to submit their interaction data to IntAct as a public repository for protein interaction data.

Reviewer #2:

Advancements in whole genome and exome sequencing have yielded rapid discovery of human missense variants which greatly outpaces current methods to functionally annotate them. Many of these variants map to intrinsically disordered regions (IDRs) which harbor most of the proteome's short linear motifs (SLiMs). SLiMs are short interaction modules 3 - 10 amino acids long which regulate key cellular processes such as localization, transactivation, complex formation and so on. However, their generally low binding affinities have made SLiM-based interactions difficult to capture in existing high-throughput protein-protein interaction assays. Previously, the authors established large-scale proteomic peptide phage display (ProP-PD) for proteome-wide discovery of motif-mediated interactions. To understand the extent to which single nucleotide variants (SNVs) can exert SLiM network rewiring, the authors expanded their ProP-PD library to include disease-associated mutations, generating 12,301 unique single nucleotide variants (SNVs) across the IDRs of 1,915 prey proteins.

The authors interrogated binding of both wild-type and mutant prey peptides against 80 bait protein domains and identified 275 mutations to modulate SLiM-based PPIs, with 367 unique domain-mutation pairs and roughly equal proportions either diminishing or enhancing interactions. By mapping mutations back to the prey motif consensus sequence within which baits bound, they found that mutations in key residues diminished interactions whereas motif-creating mutations enabled more interactions. Frequently altered interactions with E3 ligases and proteins involved in trafficking and scaffolding point to dysregulation of protein abundance and localization, respectively, as common features underlying disease progression.

Overall, the rationale for the study is sound, the experiments are well conducted, and the authors' conclusions are generally supported by the data. The authors are careful not to overstate the significance of their findings, which I find refreshing! The study will be of high interest to many scientists working on variant annotation, network biology, protein/protein interactions, and rare diseases. I think the study is suitable for publication in MSB but I suggest the authors address the points raised below. They should be relatively easy to address.

Major points

1. Phage display is an incredibly powerful assay for large-scale studies, but one of the main downsides is that it is purely an in vitro assay with purified proteins and phage-displayed peptides. Inevitably, there is always an underlying question of how the results translate to cells. The authors are careful with their interpretation and validate two interaction-disrupting or interaction-diminishing mutations in cells.

I commend the authors for including their negative results with interaction-enabling mutations, since it would have been easy to just leave those negative results out (which I suspect happens too often in proteomics validation studies). It is indeed possible, as the authors state, that the interactions may not be strong enough in the cellular context. Alternatively, the disordered region may not be easily accessible in living cells. However, I think it would be still worth it to try alternative assays with some of them. For example, increased binding to MAP3LC3B could increase autophagy-dependent turnover of BUB1 S492F, which could be assessed by western blotting after cycloheximide treatment +/- autophagy inhibition. For CTNNB1 S33F, it would be interesting to know if the mutant localizes to G3BP1-containing stress granules. This could be done by treating cells with arsenite prior to assessing CTNNB1 localization by immunofluorescence or using GFP fusions.

If indeed it turns out that no interaction-enabling mutations can be validated with full-length proteins, the authors should be cautious about wording throughout the manuscript. The concept of interaction-enabling mutations is a good one here. But for

example, the last sentence of the abstract ("The study provides a panoramic view of how disease-associated mutations perturb and rewire the motif-based interactome.") is rather strong given that not a single variant potentially increasing existing interactions or creating a novel interactions was validated.

2. Were the two interaction-disrupting interactions and three interaction-enhancing interactions the only ones the authors attempted to validate?

3. I am not sure if I understand the analysis that was done for Figure 3B. The authors compare Mann-Whitney p-values for each category and derive another p-value from one-way ANOVA. Is it not possible to compare the actual values in each category instead of p-values? For example, values could be the fraction of variants whose interaction is affected by mutations in the key residue, flanking residues, or wildcard positions. Or alternatively, the mutation enrichment score as in Figure 3A. The same comment applies to Figure 3C with Grantham scores. Perhaps I do not understand the analysis correctly, but at least the authors should explain their rationale more clearly.

4. "Of the PPIs, 106 have been previously reported in public databases and out of those 36 were also found in our previous study". Is this overlap statistically significant?

5. "Finally, we note that relaxing the p-value cut-off to p-value {less than or equal to} 0.01 would increase the numbers to 854 domain-mutation pairs affected by mutation with half being promoted (429) and half being diminished (425) by the mutation." Another way to define an appropriate threshold could be by using a benchmark set and quantifying precision/recall against the benchmark at different score thresholds.

6. Were the benign variants equally disruptive of interactions as pathogenic variants were?

7. How large were the peptides? What was the tiling window size?

8. What determined the selection of bait proteins?

9. Finally, the authors discuss the possibility of drugging motif-based interactions, and it is indeed an exciting avenue - although very challenging. In the discussion, the authors could also mention molecular glues that can modulate existing protein/protein interaction interfaces or create entirely new ones. A great example is a recent study showing that the motif-based interaction between mutant beta-catenin and beta-TrCP E3 ligase can be stabilized with a small molecule (PMID 30926793).

Minor comments

1. Fig 1A: Single nucleotide variances -> Single nucleotide variants

2. In Figure 2B, instead of (or in addition to) pie graphs it would be useful to show the relative fraction of each category in the original pool, in domain-region interactions, and in perturbed interaction pairs. The same comment applies to Figure 4A, as it is difficult to compare two pie charts to each other (especially with so many categories).

3. Many plots have $\log_{10}(p)$ as the Y axis, but how the p-value was derived is not indicated in the plot.

4. When you first reference the GenVar library, it is not explicitly introduced. It would be good to state that the "novel phage library" is the GenVar_HD2.

5. I'm wondering if the statement that 22% of disease-associated missense mutations map to IDRs is still valid, given that the original estimates were generated prior to resources like ClinVar and gnomAD. Are there more recent estimates?

Reviewer #3:

In this study, the authors explore the role of mutations in intrinsically disordered regions on short linear motif-dependent protein-protein interactions. They apply a previously utilized phage-display prey-bait methodology to this problem, and demonstrate clear effects of many disease causing mutations (somatic and germline). They confirm their findings in these short-sequence phage screens with select intact protein binding measurements, as well as in cell function assays.

This is an excellent study that adds to the growing body of knowledge regarding the effects of disease-causing genetic variants on protein-protein interactions and their functional consequences. The proteomics and genetics communities will be most interested in these results, and will likely use them to expand our understanding of protein-protein interactome function in health and disease, as well as to identify potential novel therapeutic targets for complex diseases.

Comments for the authors follow:

--On page 3, paragraph 2, please comment on the frequency of disease-causing mutations localized to protein-protein binding

interface sites vs. non-interface sites for somatic and germline mutations.

--How does the degree of the mutant protein in the PPI affect the importance (likelihood) of short linear motif-dependent binding variations on pathogenicity?

--For proteins with degree greater than 1, how does a short linear motif mutation that affects one binding interaction affect other binding interactions (if at all)? Are there allosteric consequences to these mutations? Addressing this question with one or two examples would be helpful.

Point-by-point reply to reviewers

Reviewer #1:

Kliche et al report on the results of a phage display screen for 80 motif-binding domains against a peptide library that was designed to cover disordered regions in human proteins that carry missense mutations that are associated with disease. Of note, the peptide library contained wild type and mutant peptides enabling a mapping of mutations that would decrease or increase binding to any of the presented 80 domains. To the best of my understanding this is the largest and also most systematic investigation of the effect of disease mutations falling in short linear motifs (SLiMs) on the binding of folded domains. The authors report on close to 400 domain-mutation pairs that significantly altered binding based on a stringent cutoff of which about half increase and the other half decreased binding. Findings were further validated in orthogonal assays using fluorescence polarization with domains and peptides as well as colP with full length proteins and microscopy. The study suggests that alterations of domain-SLiM mediated protein interactions might contribute to the manifestation of a diverse set of genetic diseases. The study also provides first ideas for possible molecular mechanisms underlying the pathogenicity of reported mutations.

Reply: We thank the reviewer for appreciating the novelty of our study.

Minor points

Comment 1: page 5, third line: the grammar of this sentence does not seem right

Reply: We have corrected this.

Comment 2: page 5, section title: reveal, not reveals

Reply: We have corrected this, thank you for spotting it.

Comment 3: page 5, second paragraph: I suggest rephrasing for clarity: We found that the peptide-containing prey proteins shared GO terms related to the GO categories subcellular localization, molecular function, and biological processes with ... (I had to read the sentence multiple times and look at the figure to understand what you were referring to.)

Reply: Thanks for pointing this out, we have rephrased the sentence.

Comment 4: page 5, second paragraph towards the end: Your statement about the different number of interactors found for G3BP1/2 in the previous screen and the one reported here makes me wonder about the sensitivity of the screens and whether you can estimate the fraction of mutations screened whose effect you simply failed to detect due to limited sensitivity of the phage display screens. If you can estimate this, it would be nice to add it to the manuscript. If you cannot estimate it, it would still be nice to briefly discuss this in the discussion section.

Reply: Good point. Based on the available knowledge of SLiM-based interactions we calculated the recall of known SLiM-based interactions from the phage selections using the

GenVar-HD2 library and found it to be 19.9%, which is very similar to the previously estimated recall of 19.5% for selections against the HD2 library. While we cannot directly calculate the fraction of mutations screened whose effect we fail to nail (simply because there is not sufficient data to use for comparison) we can estimate that we will miss 80% of the interactions, and hence likely in the order of 80% of the cases for which the mutations have an effect. However, this is a rough estimate as some interactions are perturbed by many mutations (that is mutational hotspots) while others are only affected by one or few mutations. We have added the recall estimate to the results (page 5), and also added the topic to the discussion (page 13).

Comment 5: page 6 section title: selection and not selections

Reply: We have corrected this. Thank you.

Comment 6: page 7, second paragraph: There are more than 100 domain-mutation pairs for which no motif consensus sequence could be identified in the peptide. Quite some of these are also shown in the little networks drawn in Fig 4c-f and Fig 5c-d. How can they be interpreted? Do they point to extensions of the known motif consensus or alternative binding modes for the domains or to false positives? It would be nice if the authors were able to do more analyses on these or at least speculate on their interpretation in the discussion section.

Reply: Thank you for raising this point. It made us realise a mistake in our analysis so that we missed annotating motifs in a number of peptides. We have now updated the analysis (See Dataset EV6) and included the motif information used for each bait protein (See Dataset EV2D-E). There are still 68 domain-mutation pairs for which there was no match with the previously annotated (or here defined) motifs. From an ocular inspection it appears as if some of the cases are missed due to overly defined motifs, or variations of the motifs and potentially the presence of variant motifs. There may of course also be some false positives. We have added a comment on this to the discussion (page 16).

Comment 7: page 8, first line of 2nd paragraph: the "that" is too much, write "which interactions are lost and ..."

Reply: We have corrected this. Thank you.

Comment 8: page 8, 2nd paragraph: The authors report at the beginning of the section that interactions functioning in the ubiquitin system are more commonly lost by disease mutations while interactions functioning in the autophagy system are more often enabled, for example. The pie charts shown in Fig 4a I thought were not very helpful to visually support this statement and it certainly does not allow for a quantification. If possible, it would be nice to compute enrichment scores to support the statements.

Reply: Thank you for raising this point. We replaced the pieplots with a back-to-back barplot (Figure 4A) that we find better illustrates the observed variations.

Comment 9: page 8, 2nd paragraph: the reference to Fig 4b misses the 4

Reply: We have corrected this. Thank you.

Comment 10: page 11, first couple of lines: Can you quantify your statement that no disease category was overrepresented in terms of mutations that showed an effect on domain-binding? Fig 6a is not very helpful to visually support this statement.

Reply: We agree that the figure did not support the expressed point in a convenient way. We therefore compiled the values and added the information to Dataset EV6B for an easy overview. The results show that there was only an increased/decreased representation of 3 disease categories after selection as compared to in the library. The peptides with mutations associated with cardiovascular/hematologic disappeared after selection, possibly because there was no domain binding to these peptides as bait in our selection. For the peptides with mutations associated with immune disease there was a minor reduction (32%) of representation after selection. In contrast, the fraction of mutations associated with reproductive disorders doubled after selections. However, in most cases, these mutations had no effect on binding as judged by the phage data, and the increase in the representation of the mutations of reproductive disorders likely has more to do with the fact that several of these peptides contain SH3 binding motifs, and we have several bait SH3 domains. Thus, the observed effect likely represents a sampling bias.

Comment 11: page 12, discussion: You state that "many of the deregulated interactions are thought to map to SLiM-based interfaces..." I would friendly disagree with this statement. I think that it is very unclear to which extent mutations in disordered regions are disease-causing because they for example fall into SLiMs. Many scientists are still uninformed about the many roles of disordered regions in proteins for protein function. As a consequence most mutations falling in disordered regions remain uncharacterized while mutations falling in domains are more readily called pathogenic (this is based on own observations and knowing how variant effect predictors work). I would suggest to reconsider this statement and potentially rephrase. This is also exactly one of the highlights of the manuscript as a first experimental study that shows the potential of deregulation of SLiM-domain interactions in disease.

Reply: We thank the reviewer for pointing this out. We have rephrased the sentence following the constructive suggestion.

Comment 12: Fig 2e: The coloring of the wild-type vs mutant title of the figure confuses me. Aren't the blue and red dots representing enrichments or depletions of mutant vs wild-type peptide reads? If I am not mistaken then I would not color the title this way or change to "disabling vs enhancing mutations". I also find the log 10 on the y-axis non-intuitive. Wouldn't a log 2 make more sense for visualization purposes?

Reply: We agree with the reviewer and have adjusted the title of the plot and the scale of the y-axis according to the comment above. We hope that the visualisation is now clearer.

Comment 13: Fig 6b and 6c: Please add to the legend an explanation of the node shape. Why did you decide to show known interactions between the preys? I would have found it more interesting to see which of the bait-prey interactions were known before.

Reply: Thank you for pointing out the lack of information. We added an explanation for the node shape (square for bait proteins, octagonal for prey proteins) to the figure legend. We

visualised the prey-prey protein interactions to provide a structured framework to the network, which we find useful. But we agree that the information on previously reported bait-prey interactions should be integrated in the network and have modified the figure accordingly (green dotted line for the previously reported interactions).

Comment 14: Fig 6d and 6e. I honestly don't find these last analyses very useful and helpful. If there is space limitations one could remove this last part.

Reply: We respectfully keep the Fig 6d and 6e as it might be of interest to some readers, and links to the potential druggability of these interactions. Reviewer 2 found this part interesting.

Comment 15: I would encourage the authors to submit their interaction data to IntAct as a public repository for protein interaction data.

Reply. We fully agree and have deposited the results in IntAct (ID: IM-30020). We have added the text "The protein interactions from this publication have been submitted to the IMEx (<http://www.imexconsortium.org>) consortium through IntAct (Del Toro et al., 2022) and assigned the identifier IM-30020" to the Data Availability section.

Reviewer #2:

Advancements in whole genome and exome sequencing have yielded rapid discovery of human missense variants which greatly outpaces current methods to functionally annotate them. Many of these variants map to intrinsically disordered regions (IDRs) which harbor most of the proteome's short linear motifs (SLiMs). SLiMs are short interaction modules 3 - 10 amino acids long which regulate key cellular processes such as localization, transactivation, complex formation and so on. However, their generally low binding affinities have made SLiM-based interactions difficult to capture in existing high-throughput protein-protein interaction assays. Previously, the authors established large-scale proteomic peptide phage display (ProP-PD) for proteome-wide discovery of motif-mediated interactions. To understand the extent to which single nucleotide variants (SNVs) can exert SLiM network rewiring, the authors expanded their ProP-PD library to include disease-associated mutations, generating 12,301 unique single nucleotide variants (SNVs) across the IDRs of 1,915 prey proteins.

The authors interrogated binding of both wild-type and mutant prey peptides against 80 bait protein domains and identified 275 mutations to modulate SLiM-based PPIs, with 367 unique domain-mutation pairs and roughly equal proportions either diminishing or enhancing interactions. By mapping mutations back to the prey motif consensus sequence within which baits bound, they found that mutations in key residues diminished interactions whereas motif-creating mutations enabled more interactions. Frequently altered interactions with E3 ligases and proteins involved in trafficking and scaffolding point to dysregulation of protein abundance and localization, respectively, as common features underlying disease progression.

Overall, the rationale for the study is sound, the experiments are well conducted, and the authors' conclusions are generally supported by the data. The authors are careful not to overstate the significance of their findings, which I find refreshing! The study will be of high interest to many scientists working on variant annotation, network biology, protein/protein interactions, and rare diseases. I think the study is suitable for publication in MSB but I suggest the authors address the points raised below. They should be relatively easy to address.

Reply: We thank the reviewer for the expressed interest in our study and for the helpful suggestions.

Major points

Comment 1. Phage display is an incredibly powerful assay for large-scale studies, but one of the main downsides is that it is purely an in vitro assay with purified proteins and phage-displayed peptides. Inevitably, there is always an underlying question of how the results translate to cells. The authors are careful with their interpretation and validate two interaction-disrupting or interaction-diminishing mutations in cells.

I commend the authors for including their negative results with interaction-enabling mutations, since it would have been easy to just leave those negative results out (which I suspect happens too often in proteomics validation studies). It is indeed possible, as the authors state, that the interactions may not be strong enough in the cellular context. Alternatively, the disordered region may not be easily accessible in living cells. However, I think it would be still worth it to try alternative assays with some of them. For example, increased binding to MAP3LC3B could increase autophagy-dependent turnover of BUB1 S492F, which could be

assessed by western blotting after cycloheximide treatment +/- autophagy inhibition. For CTNNB1 S33F, it would be interesting to know if the mutant localizes to G3BP1-containing stress granules. This could be done by treating cells with arsenite prior to assessing CTNNB1 localization by immunofluorescence or using GFP fusions.

If indeed it turns out that no interaction-enabling mutations can be validated with full-length proteins, the authors should be cautious about wording throughout the manuscript. The concept of interaction-enabling mutations is a good one here. But for example, the last sentence of the abstract ("The study provides a panoramic view of how disease-associated mutations perturb and rewire the motif-based interactome.") is rather strong given that not a single variant potentially increasing existing interactions or creating a novel interactions was validated.

Reply: We performed the experiments suggested by the reviewer. For the CTNNB1 experiment, we transiently transfected HeLa cells with either YFP-CTNNB1 wild-type, YFP-CTNNB1 S33F mutant or YFP-G3BP1 and performed live cell microscopy of the cells in response to 0.5 mM arsenite treatment. While YFP-G3BP1 localises nicely to stress granules upon treatment and as previously reported (PMID: 38177924), neither YFP-CTNNB1 wild-type nor YFP-CTNNB1 S33F mutant changed their localisation (Figure A). The experiment was performed in a single biological repeat. The expression of the CTNNB1 constructs was challenging, as observed previously, but we observed the same behaviour in at least ten cells per construct (wild-type: ca. 10 cells, mutant: ca. 20 cells). We hence conclude that arsenite treatment does not result in the recruitment of CTNNB1 wild-type or mutant to stress granules.

Figure A: Transiently transfected HeLa cells (YFP-G3BP1, YFP-CTNNB1 wild-type and YFP-CTNNB1 mutant) before (0 min) and after (30 min) treatment with 0.5 mM arsenite.

For the BUB1 experiment, we used HeLa cell lines stably expressing Flag-BUB1 wild-type or Flag-BUB1 mutant and performed a cycloheximide (300 µg/mL) chase experiment either with or without preceding autophagy blockage by bafilomycin A (100 nM) for ca. 20 h. Cells were collected at time point 0 min, 30 min, 60 min, 120 min and 240 min of the chase experiment. The experiment was evaluated by immunoblotting for endogenous GAPDH, endogenous CDC20 and the stably expressed Flag-BUB1 constructs (Figure B). First, we judge that the chase experiment has worked due to the decreasing levels of CDC20 with time in the cell, which were not treated with bafilomycin A. The chase of the BUB1 constructs is less obvious, presumably due to the stable expression of the BUB1 constructs, but quantification of the blot demonstrates the degradation at 240 min (Figure C). In a previous experiment, we tried to extend the chase to 360 min but the cells started to die at that time point, particularly when treated with bafilomycin A. From the quantification, it can be suggested that the BUB1 S492F mutant is degraded slightly faster and more linearly than the BUB1 wild-type, however that is irrespective of the bafilomycin A treatment (Figure B). From that, we conclude that the potential differential degradation behaviour is independent of the autophagic pathway. The experiment was performed as a single repeat.

Figure B: Blot of the cycloheximide chase experiment of the BUB1 wild-type and S492 mutant with and without prior Bafilomycin A treatment.

Figure C: Quantification of the cycloheximide chase experiment with the stable Flag-BUB1 wild-type and mutant cell lines, with and without Bafilomycin A treatment.

Following the suggestion of the reviewer we have softened the last sentence of the abstract.

Comment 2: Were the two interaction-disrupting interactions and three interaction-enhancing interactions the only ones the authors attempted to validate?

Reply: We tested the effect of one more mutation on the interaction in the context of the full-length proteins, namely the effect of the P348L SQSTM1 mutation on the binding to MAP1LC3B (ATG8). It is a known interaction (PMID: 17580304), which was suggested by the GenVar selection results to be enhanced by the mutation. There is also one study in support of the observation, where they stated based on a co-IP experiment: "The affinity of p62^{P348L} to LC3-II was also higher" (PMID: 31362587). However, we found in affinity measurements that the mutation had no impact on binding (Appendix Figure S5) and in the co-IPs we had inconsistent results between the triplicates. Due to these conflicting results with the literature and our own experiments, we decided to not show the blotting results (the affinity experiments are included in the manuscript) but we can provide them if wished for.

Comment 3. I am not sure if I understand the analysis that was done for Figure 3B. The authors compare Mann-Whitney p-values for each category and derive another p-value from one-way ANOVA. Is it not possible to compare the actual values in each category instead of p-values? For example, values could be the fraction of variants whose interaction is affected by mutations in the key residue, flanking residues, or wildcard positions. Or alternatively, the mutation enrichment score as in Figure 3A. The same comment applies to Figure 3C with Grantham scores. Perhaps I do not understand the analysis correctly, but at least the authors should explain their rationale more clearly.

Reply: After considering the reviewers comment we agree that the conducted analysis was sub-optimal. We also found errors in our motif annotation pipeline based on feedback from Reviewer 1, an issue we have now solved. We updated the data in Dataset EV6 and re-built Figure 3 to improve visualisation. In the new panel 3B we included an enrichment analysis that shows the effect of mutations on the interactions (diminishing/enhancing) depending on if the mutation alters an existing motif consensus (if any), or if the mutation creates a new consensus. The new panel 3C is a re-build of the previous panel D but with fixed annotations that now focuses only on those mutations that modulate the interactions (diminishing/enhancing).

Comment: 4. "Of the PPIs, 106 have been previously reported in public databases and out of those 36 were also found in our previous study". Is this overlap statistically significant?

Reply: Yes, the overlap is significant. To establish this, we first, we updated our PPI datasets for human proteome (December-2023). Our PPI datasets include experimentally validated protein-protein interactions obtained from the IntAct, HIPPIE, HURI, Bioplex and STRING databases. Next, we calculated the total number of possible unique protein-protein pairs, excluding self-binding, considering all baits used in the study and all preys in the GenVar library design (total: 147147 PPIs). Afterwards, we annotated how many of those protein-protein pairs have been characterised in our PPI dataset (known PPIs: 4638). We observed

that from 147147 possible protein-protein pairs, 4638 have been reported as interacting proteins in databases, and we used it as a background in our comparison to our selection results. In the current study, we observed 1229 possible unique high-/medium-confidence interactions at the protein-protein level, from which 100 were previously reported as experimentally validated PPIs in the aforementioned databases. Based on observed values in this study (100 known out of 1229) compared to the background library settings (4638 known out of 147147), we obtained an enrichment of 2.58 (p-value < 1×10^{-20} , hypergeometric test).

Comment 5. "Finally, we note that relaxing the p-value cut-off to p-value {less than or equal to} 0.01 would increase the numbers to 854 domain-mutation pairs affected by mutation with half being promoted (429) and half being diminished (425) by the mutation." Another way to define an appropriate threshold could be by using a benchmark set and quantifying precision/recall against the benchmark at different score thresholds.

Reply: That is true. We have previously used a benchmark set to define the criteria for differentiating between true positive hits (that is, binding peptides) and background noise. However, for the analysis we are doing here, that is mapping the effects of the mutations on binding, there is no appropriate benchmarking set, and we thus resorted to the presented strategy and the experimental validations.

Comment 6. Were the benign variants equally disruptive of interactions as pathogenic variants were?

Reply: That is an interesting question. However, our library comprises a much higher fraction of pathogenic than benign mutations (compare Figure 1C), so that the read-out is biased towards finding interactions perturbed by pathogenic mutations. Our selection results report on 1347 individual mutations, of which we find 275 to modulate 367 domain-mutation interactions. These can be mapped back to their clinical categorisation (benign, conflicting interpretation, pathogenic, no information available). The following table illustrates the retrieved clinical information in the library, for the total interactions and. modulated interactions.

Clinical overview	Library	Total interactions	Modulating mutations	Disruptive mutations	Enhancing mutations
Benign	493 (4.0%)	73 (5.9%)	18 (25% of total interactions of peptides with benign mutations)	10 (5.7%)	18 (9.3%)
Conflicting interpretation	47 (0.4%)	5 (0.4%)	-	-	-
Pathogenic	11291 (91.7%)	1217 (90.3%)	252 (21% of total interactions of peptides with pathogenic mutations)	159 (91.4%)	175 (90.6%)
No information available	470 (3.8%)	46 (3.4%)	5 (11% of interaction of peptides with no information)	5 (2.8%)	-

We conclude based on our yet limited data the benign and pathogenic mutations are equally likely to have a disruptive effect on the interactions observed with the test set of bait proteins used.

Comment 7: How large were the peptides? What was the tiling window size?

Reply: The peptides are 16 amino acids long with a 12 amino acid overlap. We have now specified this in the method section.

Comment 8. What determined the selection of bait proteins?

Reply: We explained the selection on page 5 (Bait collection and phage selections), but are happy to clarify here too. The rationale for the bait domains was guided first to include domains of proteins, which have been reported to be implicated in cancer (cancer-associated proteins) based on two previously reported studies (PMID: 34591613, PMID: 33558758). Secondly, we included domains the interactions of which were reported to be disrupted by genetic variation, equally based on a previous report (PMID: 31515488). Lastly, we aimed to cover a variety of (known) peptide-binding domains, which were partially screened previously against other phage libraries (HD2 and/or HD2_PM). The rationale for this is that mutational phage display is greatly facilitated by previous knowledge of binding motifs.

Comment 9. Finally, the authors discuss the possibility of drugging motif-based interactions, and it is indeed an exciting avenue - although very challenging. In the discussion, the authors could also mention molecular glues that can modulate existing protein/protein interaction interfaces or create entirely new ones. A great example is a recent study showing that the motif-based interaction between mutant beta-catenin and beta-TrCP E3 ligase can be stabilized with a small molecule (PMID 30926793).

Reply: We agree, and we have expanded the discussion in this direction.

Minor comments

Comment 10. Fig 1A: Single nucleotide variances -> Single nucleotide variants

Reply: Thank you for pointing this out, it has now been corrected.

Comment 11. In Figure 2B, instead of (or in addition to) pie graphs it would be useful to show the relative fraction of each category in the original pool, in domain-region interactions, and in perturbed interaction pairs. The same comment applies to Figure 4A, as it is difficult to compare two pie charts to each other (especially with so many categories).

Reply: We agree that it was difficult to compare the data in 2B and we have therefore improved the clarity and added the percentages. However, it is not possible to add a comparison to the original pool, as we do not know how many binding sites are available in the library. We have also changed the layout of Figure 4A.

Comment 12: Many plots have $\log_{10}(p)$ as the Y axis, but how the p-value was derived is not indicated in the plot.

Reply: Thank you for pointing this out. The y-axes of the plots have now been modified to indicate the statistical test (Mann-Whitney test) from which the p-values have been derived (Figure 2, 3 and 4).

Comment 13: When you first reference the GenVar library, it is not explicitly introduced. It would be good to state that the "novel phage library" is the GenVar_HD2.

Reply: This is correct, and the text has been modified accordingly.

Comment 14: I'm wondering if the statement that 22% of disease-associated missense mutations map to IDRs is still valid, given that the original estimates were generated prior to resources like ClinVar and gnomAD. Are there more recent estimates?

Reply: The number of about 20% disease-associated mutations mapping to the IDRs seems to remain accurate. For example, a more recent study showed that about 20% of cancer drivers are mutated in the IDRs (PMID: 33806614). We have added this reference to the manuscript.

Reviewer #3:

In this study, the authors explore the role of mutations in intrinsically disordered regions on short linear motif-dependent protein-protein interactions. They apply a previously utilized phage-display prey-bait methodology to this problem, and demonstrate clear effects of many disease causing mutations (somatic and germline). They confirm their findings in these short-sequence phage screens with select intact protein binding measurements, as well as in cell function assays. This is an excellent study that adds to the growing body of knowledge regarding the effects of disease-causing genetic variants on protein-protein interactions and their functional consequences. The proteomics and genetics communities will be most interested in these results, and will likely use them to expand our understanding of protein-protein interactome function in health and disease, as well as to identify potential novel therapeutic targets for complex diseases.

Reply: Thank you very much for appreciating our study.

Comments for the authors follow:

Comment 1: On page 3, paragraph 2, please comment on the frequency of disease-causing mutations localized to protein-protein binding interface sites vs. non-interface sites for somatic and germline mutations.

Reply: Good point. We have added the information that disease-causing somatic and germline mutations more frequently are localised to protein-protein binding interfaces than to non-interface sites, as elegantly shown in the cited reference.

Comment 2: How does the degree of the mutant protein in the PPI affect the importance (likelihood) of short linear motif-dependent binding variations on pathogenicity?

Reply:

The acquisition bias caused by screening against a fixed collection of protein domain makes it difficult to answer the question. However, if we focus on our generated PPI network we see that over half of the mutation containing proteins (preys) have a degree of 1:

When looking at the proportion of the mutations for each prey that we found were modulating an interaction (that is diminishing or enhancing it) there is no correlation between the degree of the prey and the proportion of interactions perturbed by mutations ($R^2 = 0.0002$). This is also true when looking at the numbers from the bait degrees perspective ($R^2 = 0.0021$).

If instead basing the analysis of the degrees of the mutant proteins on information available in publicly available databases, the degrees are higher (preys mean degree = 118, preys median degree = 90) but the correlations are still low (prey $R^2 = 0.0004$; bait $R^2 = 0.0005$). Thus

Finally, it can be noted that our library design was highly biased towards disease-associated mutations (see reply to Reviewer 2, comment 6), but that we did not see any differences in terms of interaction perturbing effects between pathogenic and benign variants.

Comment 3: For proteins with degree greater than 1, how does a short linear motif mutation that affects one binding interaction affect other binding interactions (if at all)? Are there allosteric consequences to these mutations? Addressing this question with one or two examples would be helpful.

Reply: This is an interesting question, which we realise we did not really address before. We have now added a section on the topic in the discussion. A single disease-associated mutation may affect binding to multiple binding partners. In the simple case, a SLiM, such as a WW domain binding PPxY motif, or a LIR binding to MAP1LC3s, may have the propensity to bind to several members of a given domain family. In such cases, a mutation will affect binding to all members of the family. SLiMs may also be densely located in the IDRs and may sometimes

even overlap. In such cases, there are several potential outcomes. The effect may be to abrogate the distinct binding events. We found for example an APC W2658L mutation that reduces the binding to both CSKP and to GABARAPL1. Alternatively, a mutation may affect binding to one SLiM, while leaving the other interaction unaffected. For example, we found a SQSTM1 P348L mutation (334-GDDDWTHLSSKEVD(P/L)STGELQSL-356) to not affect the binding of the LIR motif (WxxL) by the MAP1LC3B ATG domain, but to confer a 25-fold decrease in affinity of the TGE motif for KEAP1 KELCH domain. Finally, a mutation may simultaneously abrogate binding to one protein, while creating a novel binding site for another protein, as demonstrated for the beta-catenin S33F mutation, which is famous for abrogating binding to β -TrCP and which we found to create a novel G3BP1/2 NTF2-like domain binding site. In many cases, the mutated motifs are located in distinct regions of the mutated protein, and consequently mutation of one binding site will not directly affect binding to the other proteins as the binding events are independent. However, even if the binding sites and interactions per se are independent, there may be changes of the interactomes of the mutant protein. For example, the loss of an NLS will lead to cytosolic localization of the protein, and hence loss of binding with nuclear interaction partners (e.g. CDC45).

Regarding allostery, we are focusing on SLiMs found in the intrinsically disordered regions and we typically don't expect allosteric consequences as this would typically be transmitted through a folded structure. However, it is possible that some of the mutated regions make self interactions with the rest of the protein, and that these potential interactions may be affected by the mutations.

We thank the reviewer for the interesting question and have added a section on the topic to the discussion.

27th May 2024

Manuscript Number: MSB-2023-11986R

Title: Proteome-scale characterisation of motif-based interactome rewiring by disease mutations

Dear Ylva,

Thank you for sending us your revised manuscript. We have now heard back from the two reviewers who were asked to evaluate your revised study. As you will see below, the reviewers are satisfied with the performed revisions and support publication. As such, I am glad to inform you that we can soon accept the study for publication, pending some editorial issues listed below.

- Our data editors have indicated that the following needs to be corrected in the figure legends:

-- The statistical test used for data analysis should be indicated in the legend of figure 5d.

-- Information related to n should be included in the legends of figures 4c-f; 5b.

-- The error bars should be defined in the legends of figures 4c-f; 5b, e, g.

- The funding information provided in the manuscript text (Acknowledgements) should match the information entered in the online submission system. Please make sure that the following information is provided both in the manuscript and the submission system: the Ollie and Elof Ericsson foundation (YI), Knut and Alice Wallenberg, Marie Skłodowska-Curie European Training Network Grant.

- The Data Availability section should be placed before the Acknowledgments.

- Please remove the 'Authors Contributions' from the manuscript. The 'Author Contributions' section is replaced by the CRediT contributor roles taxonomy to specify the contributions of each author in the journal submission system. Please use the free text box in the 'author information' section of the online submission system to provide more detailed descriptions if needed (e.g., 'X provided intracellular Ca⁺⁺ measurements in fig Y').

- The following callouts are missing the "S" and need to be corrected: Appendix Figure 5 and Appendix Figure 7B (i.e. they should read Appendix Figure S5 and Appendix Figure S7B).

- Please provide one zip folder per figure for the Source Data. i.e. one zip folder for Figure 2, one for Figure 4 and one for Figure 5).

- The Materials and Methods section should be renamed to "Methods".

- The synopsis image is rather large and detailed and not all labels display well at the final size required. Please resupply the image as a jpg or png at the required final size (it needs to be exactly 550 px wide, and the height ideally < 500 px), ensuring that all labels are legible. We would recommend reducing the amount of text on the figure as much as possible.

- Please provide a "standfirst text" summarizing the study in one or two sentences (approximately 250 characters) and three to four "bullet points" highlighting the main findings.

- Our data integrity analyst has detected an image re-use between Figure 5H and Appendix Figure S8. We would ask you to indicate the image reuse in the respective figure legends for transparency.

They also noted that in Appendix Figure S7B the first two WB for YFP-tagged MAP1LC3B look identical, whereas in the figure legends it is stated that they represent biological replicates. Please clarify and correct this.

Please resubmit your revised manuscript online, with a covering letter listing amendments and responses to each point raised by the referees. Please resubmit the paper ****within two weeks**** and ideally as soon as possible. If we do not receive the revised manuscript within this time period, the file might be closed and any subsequent resubmission would be treated as a new manuscript. Please use the Manuscript Number (above) in all correspondence.

Click on the link below to submit your revised paper.

Kind regards,

Maria

Maria Polychronidou, PhD
Senior Editor
Molecular Systems Biology

If you do choose to resubmit, please click on the link below to submit the revision online before 11th Jun 2024.

IMPORTANT: Please note that corresponding authors are required to supply an ORCID ID for their name upon submission of a revised manuscript (EMBO Press signed a joint statement to encourage ORCID adoption).
(<https://www.embopress.org/page/journal/17444292/authorguide#editorialprocess>)
Currently, our records indicate that the ORCID for your account is 0000-0002-7081-3846.

Please click the link below to modify this ORCID:
Link Not Available

*** PLEASE NOTE *** As part of the EMBO Press transparent editorial process initiative (see our Editorial at <https://dx.doi.org/10.1038/msb.2010.72> , Molecular Systems Biology will publish online a Review Process File to accompany accepted manuscripts. When preparing your letter of response, please be aware that in the event of acceptance, your cover letter/point-by-point document will be included as part of this File, which will be available to the scientific community. More information about this initiative is available in our Instructions to Authors. If you have any questions about this initiative, please contact the editorial office (msb@embo.org).

Reviewer #1:

My comments have been fully addressed.

Reviewer #2:

The authors have further improved the manuscript and addressed all my (minor) points very well. I would like to congratulate the authors on an interesting study that will be a very useful resource for many people!

All editorial and formatting issues were resolved by the authors.

28th Jun 2024

Manuscript number: MSB-2023-11986RR

Title: Proteome-scale characterisation of motif-based interactome rewiring by disease mutations

Dear Dr Ivarsson,

Thank you again for sending us your revised manuscript. We are now satisfied with the modifications made and I am pleased to inform you that your paper has been accepted for publication.

Yours sincerely,

Sincerely,

Poonam Bheda, PhD
Scientific Editor
Molecular Systems Biology
